# ATPγS substantially defeats the biasing mechanism for kinesin steps

Vishakha Karnawat ⓘ , Algirdas Toleikis, Nicholas J. Carter, Justin E. Molloy & Robert A. Cross ⓘ ✉

Kinesin-1 microtubule motors are ATP-fuelled, twin-headed cargo transporters that step processively along microtubules, with a load-dependent directional bias. Here we show using single molecule optical trapping that 1 mM ATPγS, a slowly-hydrolysed analogue, substantially defeats the biasing mechanism, whereas 1 μM ATPγS supports it. Our data argue that nucleotide binding puts kinesin into a previously unrecognised Await-Isomerisation (AI) state that is overpopulated by ATPγS and generates slow backsteps. In the working model we propose, exit from this AI state establishes hydrolytic competence and is potentiated by load-dependent neck-linker docking, which steers the tethered head towards its next on-axis binding site. By overpopulating the AI state, ATPγS reveals its pivotal role in the biasing mechanism, whose control logic maximises forward stepping under load in ATP by coupling steered diffusion-to-capture of the leading kinesin head to load-dependent neck linker docking and nucleotide hydrolysis on the trailing head.

Kinesins are microtubule-based cargo transporters whose activities are central to the self-organisation[1] of eukaryotic cells and organisms. The archetype of the kinesin superfamily is kinesin-1. Kinesin-1 motors haul cargo in 8 nm steps towards the plus ends of microtubules[2], using a walking action in which the twin heads of each molecule interact alternately with the microtubule[3]. This alternate-heads stepping mechanism allows single kinesin-1 molecules to make headway against loads exceeding 7 pN[4,5]. Certain features of the mechanism of single-molecule stepping under load are firmly established. Whilst waiting for ATP to bind, kinesin-1 adopts a one-head-bound (1HB) waiting state in which one of its twin heads is in an apo state and strongly bound to the microtubule, whilst the other (tethered) head is in an ADP state and is prevented from binding to the microtubule[6–8]. There is evidence that in this ATP-waiting state, the tethered head is also prevented from binding unpolymerised tubulin[9]. ATP binding to the microtubule-bound head then frees the tethered head to undergo steered diffusion-to-capture and coupled MT-activated ADP release[7]. Meanwhile, the trail head hydrolyses ATP and releases Pi, converting it to a weak-binding K.ADP intermediate that unbinds from the MT and is recovered[10,11], re-establishing the 1HB ATP-waiting state. For kinesin-1 under load, we have proposed that directional stepping emerges from a competition between tight-coupled forward steps and loose-coupled

backslips[12]. For kinesin-1, we define stall force as the force at which the probabilities of forward steps and backsteps are equal[4,5]. The biasing mechanism is the mechanism that causes forward stepping to be favoured at substall loads. Whilst the evidence for these features of the kinesin-1 mechanism appears firm, other aspects remain controversial, with recent work revisiting the incidence of fast backsteps[13,14], substeps[14], sidesteps[15] and inchworming[16].

The mechanical cycle of kinesin stepping is driven by a conformational cycle of the active site. There is firm evidence for 3 chemical kinetic states of the active site, corresponding to 3 conformational states of the head, with distinct properties. In the OPEN state, the active site is empty. In the CLOSED state, the active site is occupied by ATP (or an analogue) and the switch 1 and switch 2 regions are closed together to provide hydrolytic competence. Finally, in the TRAPPED state, ADP is trapped in the active site. The OPEN and CLOSED states bind strongly to MTs and the TRAPPED state binds weakly[17]. At zero load in ATP, it is thought that ATP binding drives the kinesin head from its OPEN state into its CLOSED state, with coupled neck linker (NL) docking and hydrolysis. One step of a kinesin dimer at low load takes only ~10 ms, encouraging this straightforward model. Under load however, the picture shifts. Undocking of the NL under load destabilises the CLOSED state and likely retards hydrolysis[18]. At

Centre for Mechanochemical Cell Biology, Warwick Medical School, University of Warwick, Coventry CV4 7AL, UK. ✉e-mail: r.a.cross@warwick.ac.uk

high load and saturating ATP, the dwell before each step typically lasts several hundred milliseconds[12]. A key question then is, during ATP-driven stepping and with the NL of the bound head undocked under load, in what state is its active site?

A potentially helpful way to deconstruct the kinesin-1 stepping mechanism under load is to change its nucleotide fuel. ATPγS is a slowly-hydrolysed analogue that is identical to ATP except that one of the oxygen atoms of its gamma phosphate is replaced with a sulphur atom. Since this sulphur atom is bulkier than an oxygen atom, it is potentially more difficult to accommodate in the active site[19]. ATPγS turnover drives slow directional gliding of MTs in motility assays[20] and slow, disorderly but directional stepping of unloaded kinesin dimers[21]. There is currently no structure for a kinesin-ATPγS complex, which is, of course, only transiently stable, but there is a structure for myosin-ATPγS, obtained using a small molecule inhibitor that blocks hydrolysis[22]. In this myosin structure, the gamma thiophosphoryl of ATPγS is rotated to allow it to project away from the catalytic centre. It is possible that a similar rearrangement is required in kinesin and that the time required slows hydrolysis. The explicitly steric action of ATPγS contrasts with the action of another widely-adopted

nonhydrolyzable analogue, AMPPNP[23]. In AMPPNP, kinesin readily adopts a CLOSED state, but hydrolysis is inhibited because in AMPPNP, the hydrolysable P-O-P linking the beta and gamma phosphates of ATP is replaced with a P-N-P linkage. The exact mechanism by which ATPγS slows kinesin-driven hydrolysis is not yet fully clear, but it is clear nonetheless that ATPγS is a very useful tool. We show here using single-molecule optical trapping that 1 mM ATPγS largely defeats the guidance mechanism that steers on-axis forward stepping of kinesin-1 under load, whilst 1 μM ATPγS does not. This differential effect provides insight into the componentry and action of the guidance mechanism.

## Results & Discussion

We first confirmed that ATPγS binds tightly, drives slow MT gliding in motility assays and competes effectively with ATP for the kinesin active site (Fig. 1a–c). In agreement with previous reports[20], the MT gliding rate in 1 mM ATPγS is 12 nm s$^{-1}$, whereas in 1 mM ATP it is ~800 nm s$^{-1}$. Single kinesin molecules in ATPγS step at 13 nm s$^{-1}$ [21]. Under load in the optical trap, ATPγS-driven single molecule stepping superficially resembles ATP-driven stepping, except that it is much slower

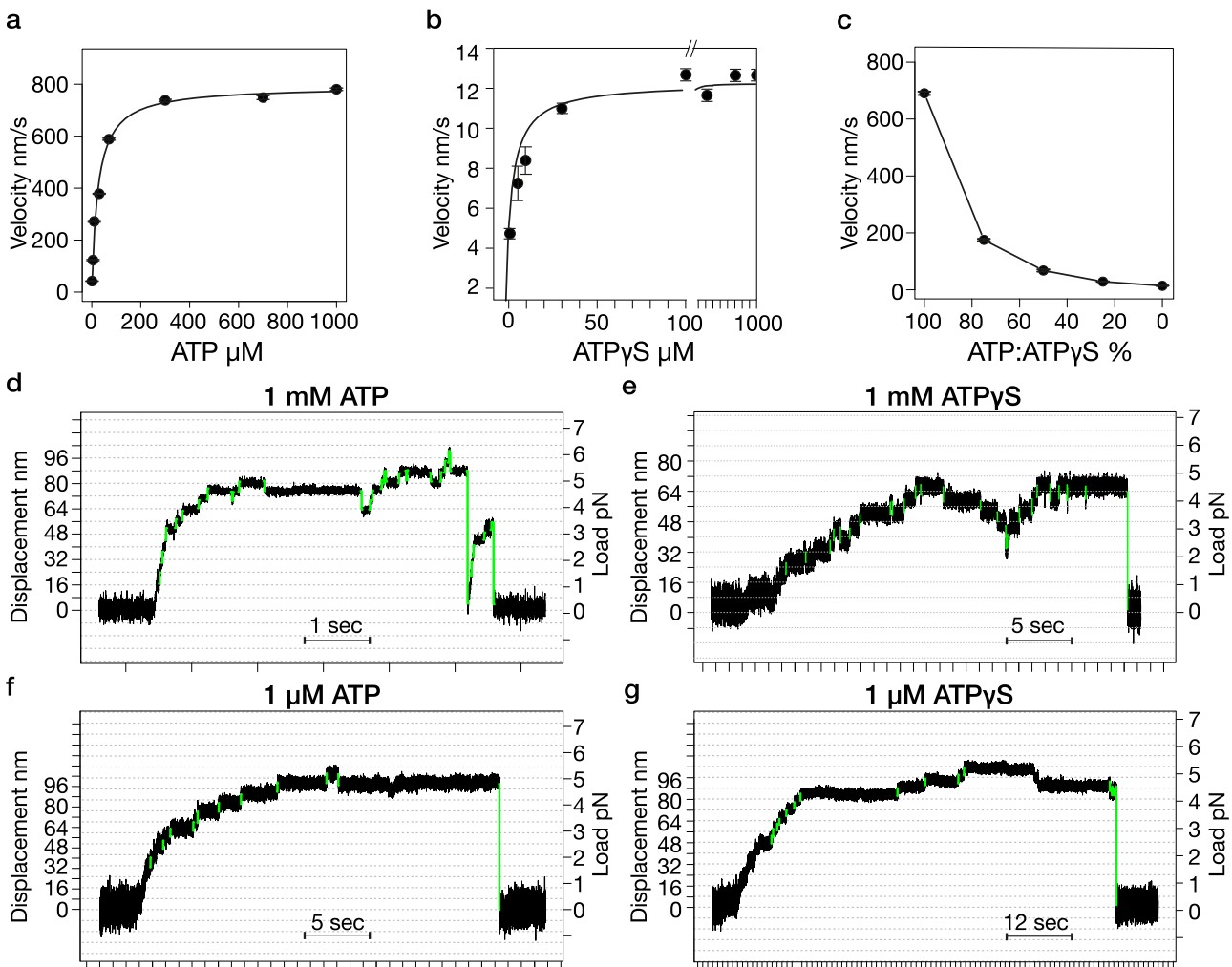

**Fig. 1 | Kinesin motility in ATP and ATPγS.** Microtubule gliding velocity in kinesin surface assays as a function of nucleotide concentration for (**a**) ATP; (**b**) ATPγS. Data are fitted with the Michaelis-Menten function: $K_m$ [ATP] = 25.7 μM, $V_{max}$ [ATP] = 790 nm/s, $K_m$ [ATPγS] = 3.1 μM and $V_{max}$ [ATPγS] = 12 nm/s. **c** Effect of the ratio of ATP to ATPγS on microtubule gliding velocity. The x-axis represents the percentage of ATP relative to ATPγS, ranging from 100% ATP (0% ATPγS) to 100% ATPγS, with the corresponding microtubule gliding velocities plotted on the y-axis. Note the

~75% reduction in gliding velocity with the addition of 25% ATPγS. *n* = 26 (**a**), *n* = 31 (**b**), *n* = 15 (**c**). Error bars show ± standard error of the mean (SEM) (**d**–**g**) example trapping records. Vertical divisions are at 1 second intervals. Horizontal grid lines are 8 nm apart. Steps marked in green are detected based on their t-score cutoff (see Methods). Step detection was performed using t-score cutoffs of 15 for **d**–**f** and 18 for **g**.

(Fig. 1d–g). Both 1 mM and 1 μM ATPγS drive kinesin to step with ATP-like processivity to a stall force that approximates that in ATP. We find that dwell times in 1 mM ATPγS are less load-dependent than those in 1 μM ATPγS, and that there are more backwards events at lower load in 1 mM ATPγS than in 1 μM ATPγS.

## 1 mM ATPγS largely defeats the biasing mechanism, but 1 μM ATPγS supports it

The forward-to-backward (F/B) probability ratio for kinesin stepping gives a measure of the directional bias. In ATP, the F/B ratio diminishes exponentially with load (Fig. 2). The exponential factors are similar at 1 μM and 1 mM ATP (Compare Fig. 2c, g, k), as noted previously for 10 μM and 1 mM ATP[4,5]. By contrast, in 1 mM ATPγS, the F/B ratio depends only very weakly on load (Fig. 2d). This is because in 1 mM ATPγS, at substall loads, the probability of 8 nm backwards displacements is increased at all loads compared to that in 1 mM ATP, at the expense of forward steps, whilst the probabilities of longer backslips or detachments are little affected (compare Fig. 2a, b). By translating the microscope piezo stage at a trigger load (see Methods), we were able to rapidly apply superstall loads and improve sampling of kinesin stepping at high loads (Fig. 2e, f). Inclusion of the extra high load data obtained in this way did not appreciably change the F/B vs. load relationships obtained without the stage-step, for either ATP or ATPγS (Fig. 2g, h). F/B step ratio measurements obtained over a range of loads thus reveal that 1 mM ATPγS largely, though not entirely, defeats the mechanism that biases kinesin-1 towards forward stepping under load. Remarkably, at low load (<2 pN), the F/B step ratio was 100:1 in 1 mM ATP but only 3:1 in 1 mM ATPγS. The F/B ratio approaches unity for both substrates at a similar value. In 1 μM ATPγS at substall loads, the F/B step ratio shows an exponential load-dependence (Fig. 2l) similar to that in 1 μM ATP (Figs. 2k) and 1 mM ATP (Fig. 2c, g), because 1 μM ATPγS generates far fewer extra 8 nm backsteps at low load than 1 mM ATPγS (compare Fig. 2j with Fig. 2b).

## Dwell times in ATP increase exponentially with load, yet are almost load-independent in 1 mM ATPγS

In search of further insight, we compared the dwell times under load for stepping in ATP and ATPγS. The dwell time is the waiting time before each step. To examine the variation of dwell times with load, we averaged all dwells within 1 pN force bins (Fig. 3). For steps in both 1 mM ATP and 1 μM ATP, the mean dwell time within each substall force bin increases exponentially with load (Fig. 3), for both forward and backward steps, reaching a plateau value at or around stall force. In ATP, average forward step dwell times at any substall load are consistently shorter than average backward step dwell times at the same load. We previously suggested that this difference arises because backwards displacements in ATP originate from a later state in the mechanochemical cycle than forward steps[12]. We proposed that backwards displacements in ATP are backslips and not backsteps and that backslips occur only after a time window for forward stepping has been closed by Pi release[12]. Our present results add measurements in 1 μM ATP and are fully consistent with this picture. ATPγS-driven stepping is, however, different. In contrast to our findings with 1 mM ATP (Fig. 3a, c), mean dwell times in 1 mM ATPγS at substall loads are almost load-independent (Fig. 3b, d). Whereas dwell times in ATP increase ~35-fold from around 13 ms at zero load (by extrapolation) to around 400 ms at stall force (Fig. 3a), in 1 mM ATPγS, dwells increase only ~2-fold from ~500 ms at zero load, to ~1000 ms at 6 pN (Fig. 3b). Stepping the stage to improve sampling at high loads did not appreciably change this picture (Fig. 3c, d).

## 1 μM ATPγS restores an exponential load-dependence of dwell times

Remarkably, reducing the concentration of ATPγS to 1 μM, well below the apparent $K_m$ of 3.1 μM measured in microtubule gliding assays

(Fig. 1b) restores an exponential load-dependence of dwell times (Fig. 3f), with an exponential factor approaching that in 1 μM ATP (Fig. 3e). In both 1 μM ATPγS and 1 mM ATPγS, forward and backstep dwell times within each force bin are not obviously different from one another (Fig. 3b, d, f).

## Cumulative probability plots clarify the load-dependence of dwell times

Averaging all the dwell times in each force bin (Fig. 3) points to an exponential dependence of dwell time on load but tells us nothing about the distribution of dwell values within each bin. To examine this point, we reanalysed the data using a cumulative probability approach. The plots obtained (Fig. 4a–d) show directly that the dwell data within each force bin are exponentially distributed. The characteristic dwell time τ, the inverse of the fitted decay constant, is obtained by single exponential fitting of all the dwell times within each bin. The data are well fit by single exponentials, and the values obtained confirm that for ATP-driven stepping, forwards and backwards dwell times increase with load according to the same exponential factor and that forward step dwells are appreciably shorter on average than backstep dwells. For ATPγS-driven stepping, dwell time distributions are also well fit by single exponentials, with a much steeper exponential dependence of dwell times on load in 1 μM ATPγS than in 1 mM ATPγS. In both 1 μM and 1 mM ATPγS, average forestep and backstep dwell times are not detectably different. Within each force bin, a few longer dwell times are inconsistent with a single exponential fit, based on their residuals (Methods). Bi-exponential fitting of the data within each force bin did not markedly improve the fits (Supplementary Figs. 1–6). One possibility is that these unusually long dwells include time spent on extra (futile) nucleotide turnovers. All the component plots and fits used to construct Fig. 4 are shown in Supplementary Figs. 1–6.

## ATP generates backslips under load, whereas ATPγS generates mostly backsteps

A further indicator of the action of ATPγS comes from the finding that whilst in ATP, backwards displacements show a spectrum of amplitudes at multiples of 8 nm (Fig. 4e), in ATPγS backwards events are almost all 8 nm (Fig. 4f). This suggests that in ATP, backwards displacements are slips, with a fixed probability of re-engagement at each backwards site[12], whereas in ATPγS, backward displacements are backsteps whose amplitudes are constrained by the combined extended length of the twin neck linker (NL) domains. Figure 4e–h show this effect for high load data, obtained by stage-stepping, for which positional noise is reduced and amplitudes are clearest. Example traces are shown in Supplementary Fig. 7. Comprehensive data on backstep amplitudes across all loads are shown in Supplementary Fig. 8.

## Added ADP provokes backslips in ATP but backsteps in ATPγS

To further interrogate the mechanics of ATPγS-driven backward displacements, we supplemented ATP or ATPγS with ADP. In 0.9 mM ATP, adding 0.1 mM ADP increases the incidence of 8 and 16 nm backslips and (especially) full detachments, at the expense of forward steps (Fig. 5a, b; compare Fig. 2a) whilst reducing the F/B ratio at each load (Fig. 5d). Adding 0.1 mM ADP also reduces the exponential dependence of dwell times on load and reduces the difference between the average forward and backstep dwell times at any particular load (Fig. 5c, e; compare Figs. 3a, 4a). By contrast, supplementing 0.9 mM ATPγS with 0.1 mM ADP specifically provokes extra 8 nm backsteps (Fig. 5g; compare Fig. 2b), whilst further reducing the already minimal load dependence of the F/B ratio (Fig. 5i) and dwell times (Fig. 5j,k), compare Figs. 3d, 4b). In trapping records, the tendency for added ADP to provoke extra 8 nm backsteps in ATPγS is clearest directly prior to the detachment events that terminate each processive run (Fig. 5a,f). In ATP plus ADP, detachments are routinely preceded by a

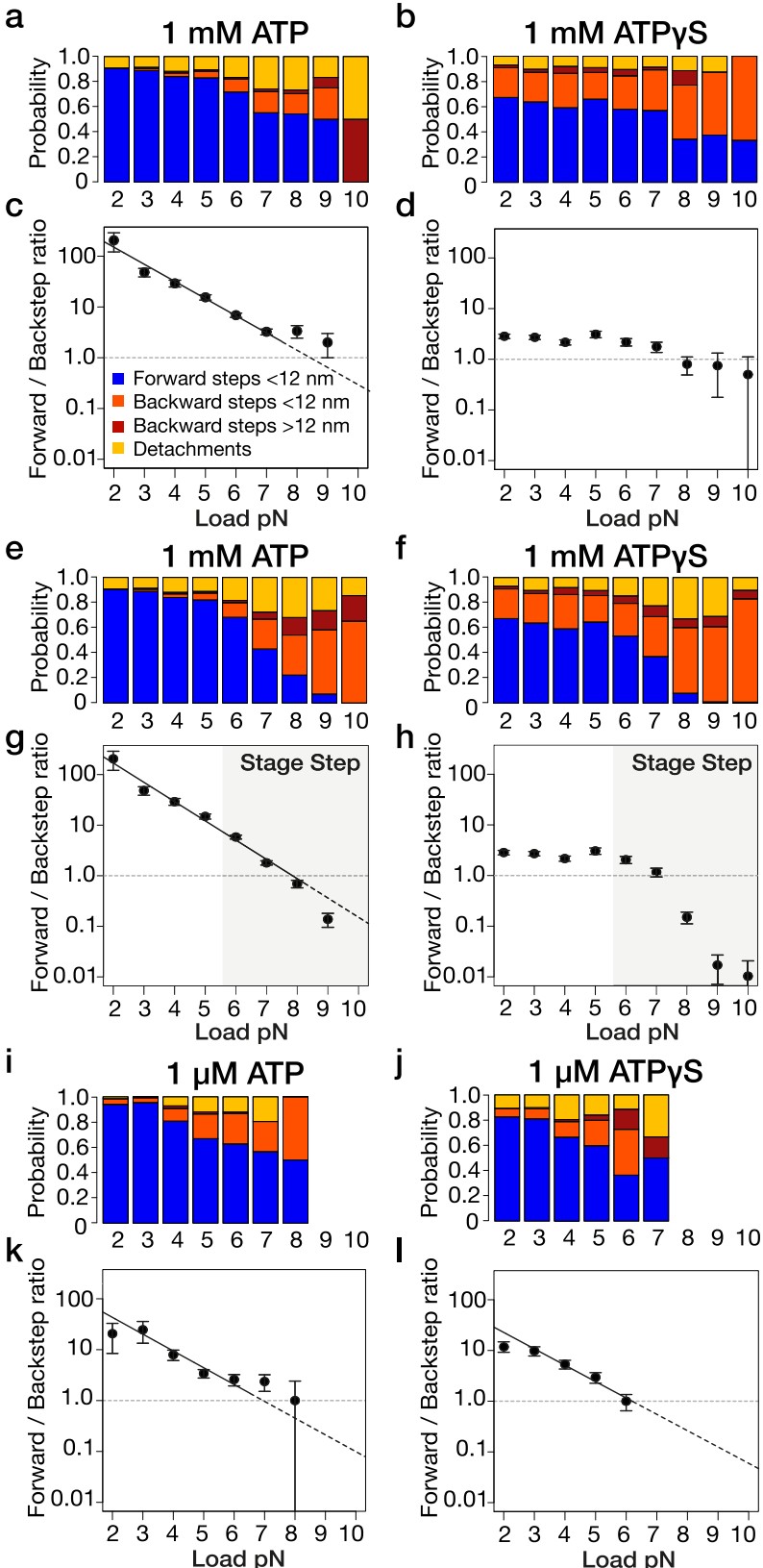

**Fig. 2 | Step-type probabilities and F/B step ratios under load in ATP versus ATPγS. a**, **b** Probability of forward steps (blue), <12 nm backsteps (orange), >12 nm backsteps (reddish-brown), and detachments (yellow) versus load for optically trapped kinesin-1 beads in (**a**) 1 mM ATP, (**b**) 1 mM ATPγS, (**i**) 1 μM ATP, and (**j**) 1 μM ATPγS. **c**, **d**, **g**, **h**, **k**, **l** F/B ratio (#forward steps / #backsteps) versus load. **c** 1 mM ATP (fitted: 2-7 pN), F/B ratio = 714 e$^{-0.78 \cdot load}$, **d** 1 mM ATPγS, **k** 1 μM ATP (fitted: 3-6 pN), F/B ratio = 197 e$^{-0.76 \cdot load}$, and **l** 1 μM ATPγS (fitted: 3-6 pN), F/B ratio = 100 e$^{-0.74 \cdot load}$. **e**–**h** equivalent to **a**–**d** except sampling of the area shaded grey was improved by stepping the stage at a trigger load to apply superstall force (Methods). **g** 1 mM ATP (fitted: 2–7 pN), F/B ratio = 977 e$^{-0.88 \cdot load}$. **h** 1 mM ATPγS. $n$ = 8356 (**c**), $n$ = 2491 (**d**), $n$ = 9144 (**g**), $n$ = 3166 (**h**), $n$ = 750 (**k**), $n$ = 1123 (**l**). Error bars show ±SEM.

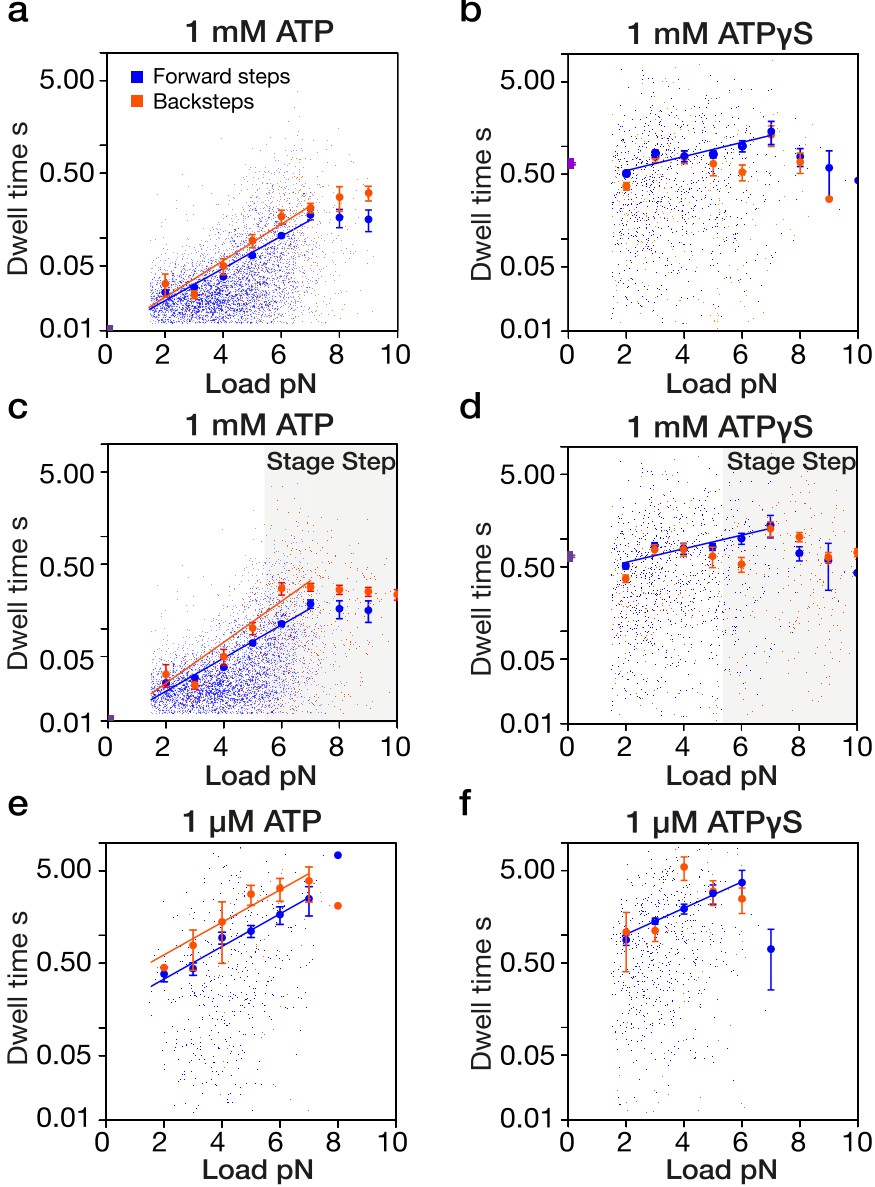

**Fig. 3 | Load-dependence of average step dwell times in ATP versus ATPγS.**
**a** Load dependence of step dwell times on load for forward steps (blue) and backward steps (orange) at 1 mM ATP (fitted: 2–7 pN). All the data are plotted. Larger symbols represent the mean dwell times calculated for 1 pN interval bins. Fits are dwell (forward) = $0.0094\ e^{0.40\ *\ load}$ and dwell (backward) = $0.0094\ e^{0.45\ *\ load}$.
**c** Same but superstall load was applied by triggering a stage-step (Methods). Fits are dwell (forward) = $0.0092\ e^{0.41\ *\ load}$ dwell (backward) = $0.0092\ e^{0.51\ *\ load}$, (fitted: 2–7 pN). **b** 1 mM ATPγS, dwell (forward) = $0.4067\ e^{0.17\ *\ load}$ (fitted: 2–7 pN). **d** Same but applying superstall load by triggering a stage-drag, dwell (forward) = $0.4135\ e^{0.16*load}$ (fitted: 2–7 pN). **e** 1 μM ATP. Fits are dwell (forward) = $0.1500\ e^{0.41\ *\ load}$ and dwell (backward) = $0.2743\ e^{0.41\ *\ load}$ (fitted: 3-7 pN). **f** 1 μM ATPγS. Fit is dwell (forward) = $0.5308\ e^{0.33*load}$ (fitted: 3–6 pN). Fitting in all cases is by least-squares to log (y) = k*x + b. Errors bars show ± SEM. $n = 8356$ (**a**), $n = 2491$ (**b**), $n = 9144$ (**c**), $n = 3166$ (**d**), $n = 750$ (**e**), $n = 1123$ (**f**).

forward step (Fig. 5l). By contrast, in ATPγS plus ADP, detachments are commonly preceded by an 8 nm backstep (Fig. 5m). ADP supplementation thus reinforces the view that the mechanics of ATPγS-driven stepping differ markedly from those in ATP. Below, we propose a working model in which supplementing ATP with ADP will predominantly increase backslipping from a 1-head-bound (1HB) dwell state, whilst supplementing 1 mM ATPγS with ADP will predominantly increase backstepping from a 2-heads-bound (2HB) dwell state. Note that these ADP supplementation experiments also act as a control for the possibility that small amounts of contaminating ADP in our stock nucleotides might account for the observed differences between ATP-driven and ATPγS-driven stepping. Adding ADP to ATP does not generate ATPγS-like mechanics, or vice versa.

## Proposed working model

Our data show that 1 mM ATPγS substantially defeats the biasing mechanism for on-axis directional stepping, by generating extra backsteps. Further, the average dwell times for ATPγS steps appear almost load-independent. How do these findings illuminate the biasing mechanism? Our proposed working model (Fig. 6a) is based on that given by Toleikis et al.[12], in which, during ATP-driven stepping, the motor proportionates between forwards step (blue), backslip (orange) and no-step (stand-in-place; yellow) pathways, depending on load. In our revised model, these pathways are accessed via an additional, hitherto-unrecognised state, the Await-Isomerisation (AI) state, which also feeds a backstep pathway (Fig. 6a, pink pathway). Backstepping (pink) is mechanically distinct from backslipping (orange): backsteps

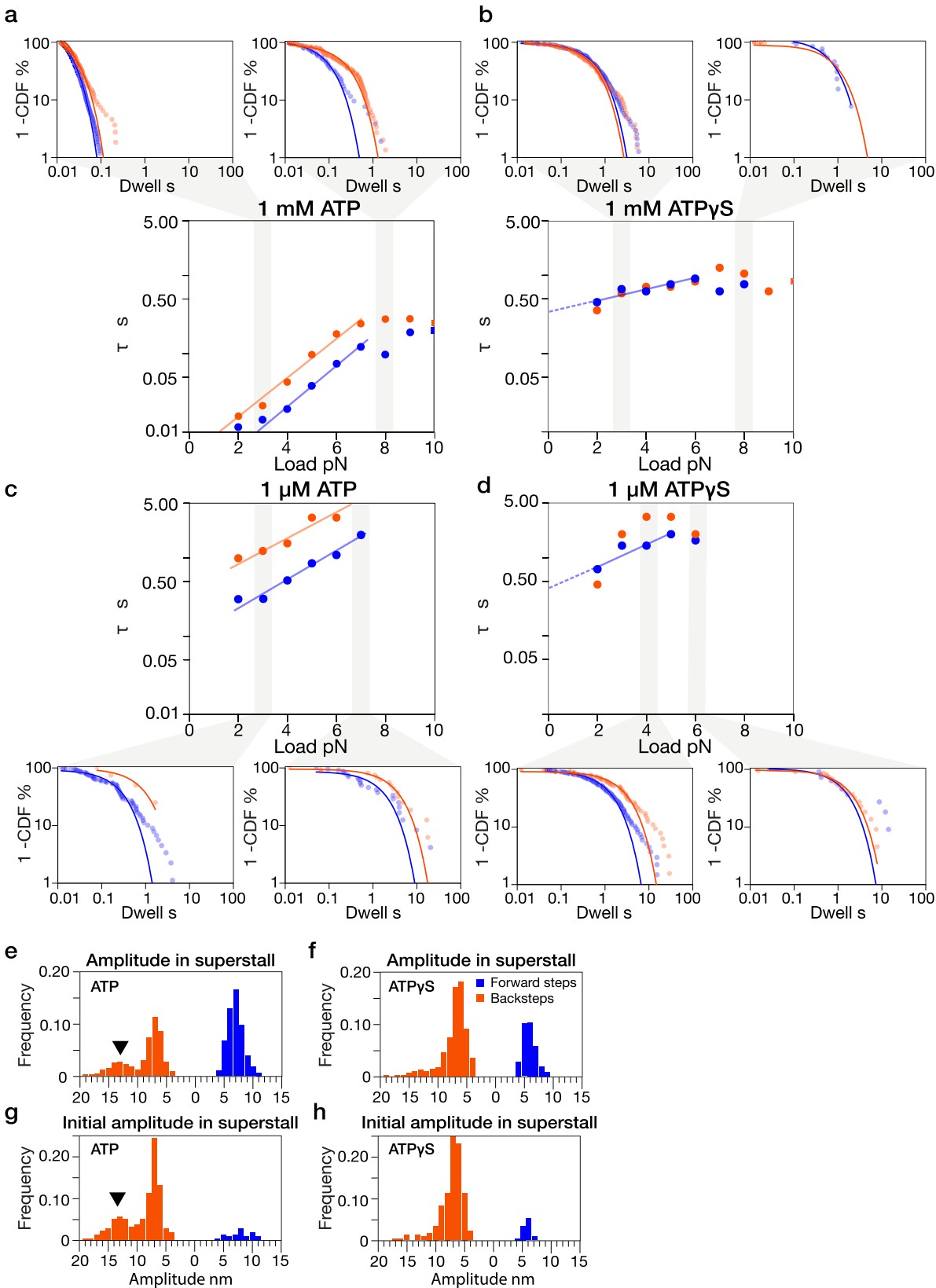

**Fig. 4 | Cumulative frequency analysis of load versus dwell time in ATP vs ATPgS. a**–**d** Normalized 1-CDF (%) plots of step dwell times for forward and backward steps across different conditions, fitted with single exponentials. **a** 1 mM ATP, cumulative distribution plots for bins 2.5–3.5 pN (left) and 7.5–8.5 pN (right), with the corresponding plot of mean dwell time ($\tau = 1/\lambda$) vs. load (middle, fitted: 4–7 pN), dwell (forward) = 0.0018 $e^{0.62 * load}$ and dwell (backward) = 0.0049 $e^{0.58 * load}$. **b** 1 mM ATPγS, $\tau$ vs. load plot (fitted: 2–6 pN) dwell (forward) = 0.3619 $e^{0.15 * load}$.

**c** 1 µM ATP, $\tau$ vs. load plot (fitted: 3-7 pN), dwell (forward) = 0.0971 $e^{0.43 * load}$ and dwell (backward) = 0.3895 $e^{0.38 * load}$. **d** 1 µM ATPγS, $\tau$ vs. load plot (fitted: 2–5 pN), dwell (forward) = 0.4430 $e^{0.31 * load}$. **e**, **f** step amplitude distributions for ATP (1 mM, $n = 814$) and ATPγS (1 mM, $n = 747$) steps obtained at superstall loads after a stage-step. **g**, **h** same but only for the initial step that followed each stage-step. $n = 213$ for **f** and $n = 233$ for **h**.

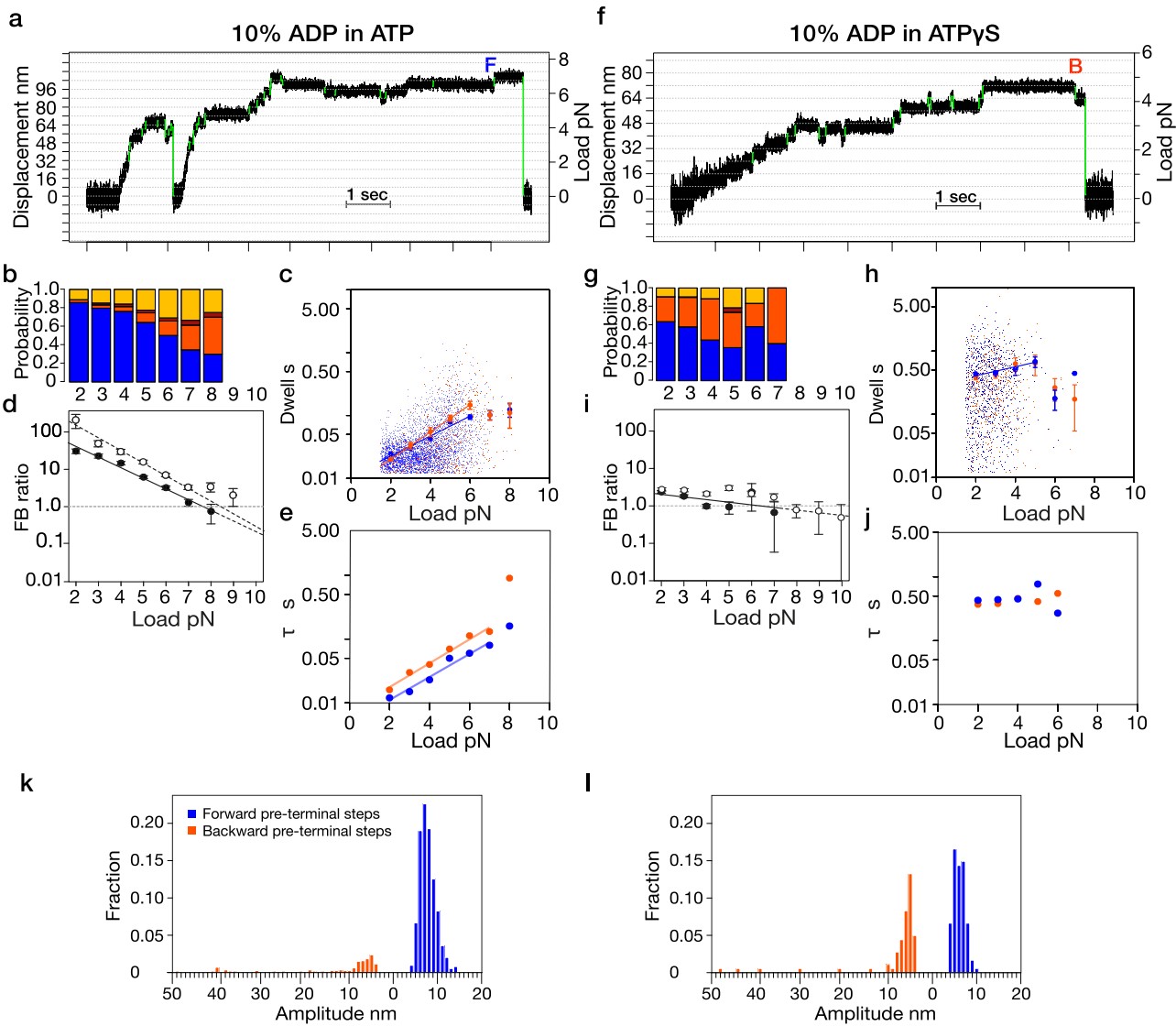

**Fig. 5 | Added ADP causes a pre-detachment 8 nm backstep in ATPγS but not in ATP.** Example trapping records for **a** 0.9 mM ATP plus 0.1 mM ADP. **b** 0.9 mM ATPγS plus 0.1 mM ADP. In ATP, detachment is preceded by a forward 8 nm step (F). In ATPγS, detachment is preceded by a backward 8 nm step (B). **b, g** Step-type probabilities (colour coded as in Fig. 2a–d). **b–j**, (**d, i**) Forward-to-backward (F/B) step ratios. **d** 10% ADP: ATP (filled circles, fitted: 2-8 pN), F/B ratio= 152.51 $e^{-0.66 * load}$, 1 mM ATP (open circles, shown for comparison). See Fig. 2C).**i** 10% ADP: ATPγS (fitted: 2–7 pN), F/B ratio = 2.81 $e^{-0.16 * load}$, 1 mM ATPγS (open circles, shown for comparison. See Fig. 2D). Fitting in all cases is by least-squares to log (y) = k*x + b.

Error bars show ± SEM. **c, e, h, j** Load dependence of step dwell times. **c** Fits are dwell (forward, blue) = 0.0109 $e^{0.37 * load}$ and dwell backward (orange) = 0.0075 $e^{0.49 * load}$ (fitted: 2-6 pN). **e** Mean dwell time (τ = 1/λ) vs. Load as determined from cumulative probability distribution. Fits are dwell (forward) =0.0049 $e^{0.41 * load}$ and dwell (backward) = 0.0076 $e^{0.43 * load}$. **h** 0.9 mM ATPγS plus 0.1 mM ADP. Fits are dwell forward = 0.2733 $e^{0.18 * load}$ (fitted: 3–7 pN). n = 8733 (**c, d**), n = 2054 (**i, h**). **k,l** Amplitude distributions for the step immediately preceding detachments in (**k**) ATP + ADP (n = 1330), **l** ATPγS + ADP (n = 181).

occur hand-over-hand, backslips via a slip and re-engage process[12]. The AI state is generated by nucleotide binding to the OPEN (apo) state of the trailing kinesin head and is defined as an ATP- or ATPγS-bound state that is activated for stepping but not for hydrolysis. The AI state has a fully undocked NL. In ATP, we envisage that the AI state is in rapid equilibrium with the CLOSED state, which has a docked or partially-docked NL and is hydrolytically competent. In ATPγS, we propose that the AI state is hyper-enriched because its isomerisation into the CLOSED state is retarded. Dwelling in the AI state can then account for both the slow turnover of ATPγS and the extra backsteps we see in 1 mM ATPγS. Once the AI state isomerises into the CLOSED state, load-dependent NL docking can stabilise the hydrolytically-competent CLOSED conformation, provoking hydrolysis whilst simultaneously steering the tethered head to its next on-axis forwards binding site.

We propose the existence of the AI state based on our mechanical evidence, but our proposal is consistent with the available kinetic and structural evidence. Concerning entry to the AI state, there is strong evidence from solution kinetics that in the absence of nucleotide, the motor maintains a 1HB ATP-waiting state, with the MT-bound head in its OPEN (apo) state and the tethered head in its TRAPPED ADP state. The binding of ATP to this ATP-waiting state then triggers 'halfsite' ADP release[7]. ATPγS binding to the MT-attached head also triggers MT-activated halfsite ADP release[24], as detected using Mant-ADP, at ~30 s[-1], much faster than the ~2 s[-1] rate of ATPγS-driven stepping[21] and the ~3 s[-1] rate of MT-activated ATPγS turnover[12] (Fig. 1). Concerning exit from the AI state, there is firm structural and biochemical evidence that NL docking stabilises the hydrolysis-competent CLOSED state of the kinesin head[17]. Disabling NL docking by mutagenesis profoundly

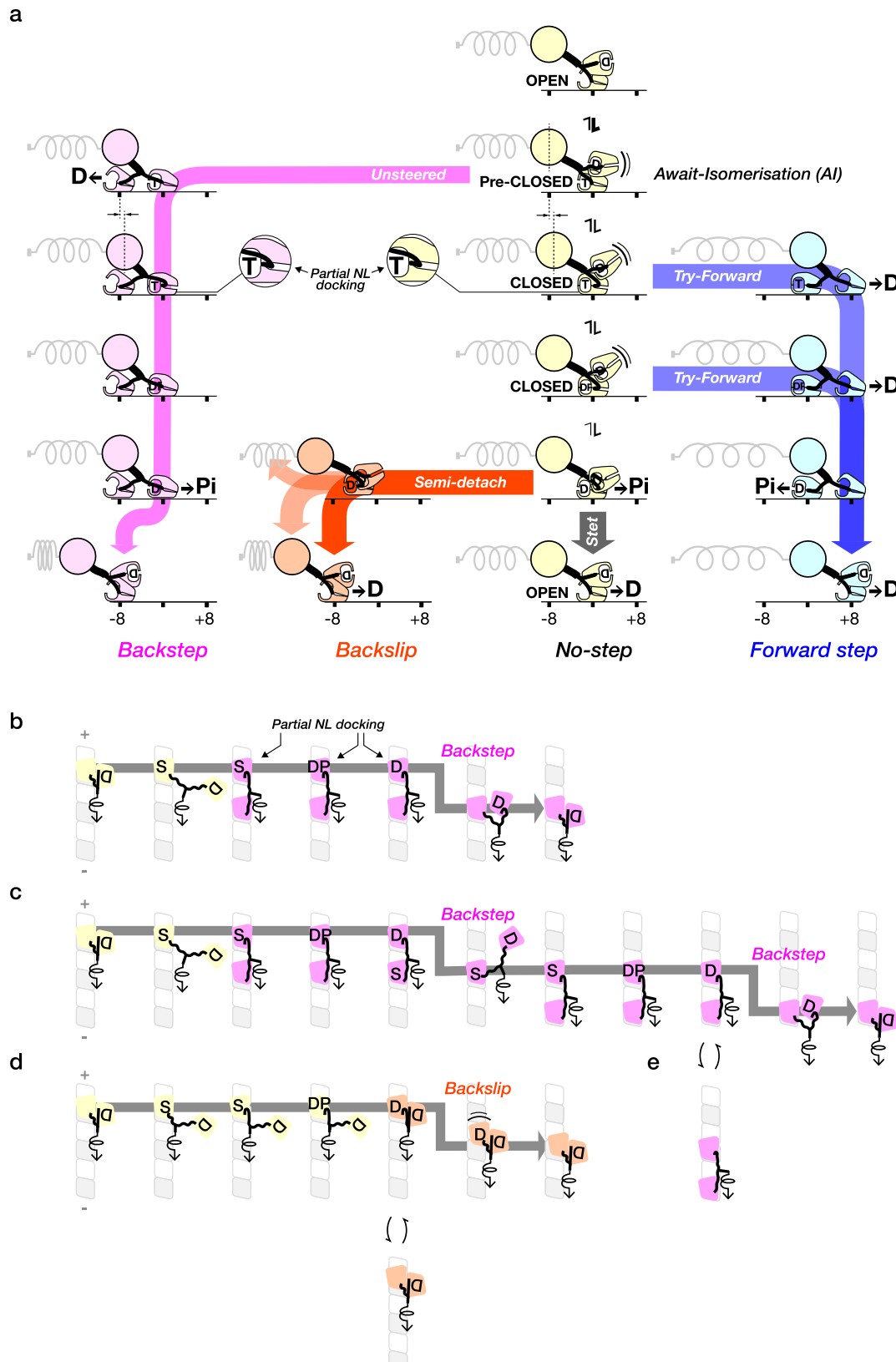

inhibits the MT-activated ATPase of kinesin-1[18,25], consistent with the CLOSED state being stabilised not only by ATP binding and MT binding[26], but also by NL docking. Recent cryoEM work[27] provides direct structural evidence that NL docking can control a subdomain motion that establishes hydrolytic competence. Further, there is

cryoEM evidence that a kinesin head with an undocked NL (the leading head of a 2HB Kif1a dimer) can bind nucleotide (AMPPNP) stably[28]. Recent data on kinesin-1[10,11] encourage this view - indeed Niitani et al. [11] argue for a stepping scheme in the absence of external load resembling that which we propose here for the loaded case.

**Fig. 6 | Proposed scheme for kinesin stepping under load ATP binding populates the Await-Isomerisation (AI) state, by shifting the trailing, MT-attached head from an OPEN into a pre-CLOSED conformation. a** This shift triggers the tethered head to begin searching for its next site. In ATP, the AI state is transient at low load, but in ATPγS, the AI state is enriched, because ATPγS retards the isomerisation from pre-CLOSED to CLOSED. The AI state is incompetent to dock the NL and so tends to generate missteps. Under load, these are mostly backsteps (pink pathway). Isomerisation of the AI state allows entry to the Try-Forward era of the cycle (blue pathway). Full y docking the NL steers the tethered head to its next on-axis forwards binding site, maximally stabilising the CLOSED state and maximally potentiating hydrolysis. Hydrolysis can also proceed via partial NL docking, as shown (callouts), at a correspondingly slower rate. Competition between this slow background rate of hydrolysis and NL-accelerated hydrolysis creates a load-dependent time window for forward stepping. At high loads, NL docking increasingly fails to occur within this time window, so that slow flux into the No-steps (yellow) and Backslips (orange) pathways comes to dominate. Untethered backslips (orange) can evolve into full detachments, but more commonly are rescued via reattachment, reverting the motor to its ATP-waiting state and allowing forward stepping to be retried, at the expense of futile cycling. **b–d** illustrate specific scenarios, viewed en face. In **b**, ATPγS (S) binding triggers a single 8 nm backstep with reversion to the ATP-waiting state. This behaviour depends on the rear head remaining in its apo state whilst the lead head is recovered and so will dominate at 1 μM ATPγS. In **c**, ATPγS binds earlier, causing a second 8 nm backstep. This behaviour will dominate at 1 mM ATPγS. **d** is an 8 nm backslip, promoted by ADP rebinding to the front head. **e** is an 8 nm backstep, promoted by ADP rebinding to the front head at low ATPγS concentrations.

## Why do dwell times depend exponentially on load?

In the scheme in Fig. 6, dwell times will be dominated by two main components, time spent waiting for ATP or ATPγS to bind (much increased in 1 μM ATP and 1 μM ATPγS) and time spent waiting for isomerisation of the AI state into the CLOSED state, with coupled NL docking. We posit that only the second of these components is load-dependent. The exponential dependence of dwell time on load (Fig. 3, Fig. 4a–d) then will directly reflect the load-sensitivity of the NL docking equilibrium. At very high loads, dwell times in both ATP and ATPγS become almost load-independent, reflecting that under these conditions, the NL almost never docks. At substall loads, the average time required to complete NL-steered diffusion-to-capture of the tethered head by its next forward binding site increases exponentially with load, consistent with tethered diffusion over an energy barrier[4].

## Why are backwards dwells longer on average than forwards dwells?

In our working model (Fig. 6), ATP binding opens a 'try-forward' time window during which the motor tries to dock the NL under load. Forward steps (blue pathway) occur when it succeeds, backslips (orange pathway) when it does not. The try-forward time window is closed by Pi release. Average backstep dwells are longer than average forward step dwells because the dwell times for backslips contain time spent before Pi release attempting but failing to step forward[12].

## Why are load vs dwell time curves different for ATP versus ATPγS?

According to our working model, nucleotide binds load-independently to create the AI state, and isomerisation of the AI state then feeds the Try-Forward pathway (Fig. 6, blue pathway). Dwell times at substall load are markedly more load-dependent in 1 mM ATP than in 1 mM ATPγS. In 1 mM ATP, dwells are ~10 ms at zero load, rising to ~500 ms at and above stall (Fig. 4a). In 1 mM ATPγS, dwells vary between ~500 ms at zero load and ~1 s at stall (Fig. 4b). Our model is consistent with this behaviour because in ATPγS, we expect an additional component of time spent dwelling in the pre-hydrolysis state. We can estimate this load-independent Await-Isomerisation time to be ~500 ms on average, for both 1 μM and 1 mM ATPγS, by extrapolating load versus dwell time plots to zero load (Figs. 3d–f, 4b, d). The average dwell time at any substall load in 1 μM and 1 mM ATPγS will then be equal to this await-isomerisation time plus the average dwell time seen in 1 μM or 1 mM ATP, because following isomerisation of the AI state into the CLOSED state, the rest of the cycle is identical to that in ATP. The effect will be to add ~500 ms to the average dwell time at all loads, thereby flattening the load versus dwell time curves (Supplementary Fig. 9).

## Why are forward and backstep dwell times different in ATP but the same in ATPγS?

In our model, ATP-driven backslips and detachments (Fig. 6, orange pathway) have longer average dwell times than ATP-driven forward steps, because they originate later in the cycle than forward steps[12]. By contrast, in ATPγS, average dwell times for forward and backwards displacements are indistinguishable (Fig. 3e). Our working model envisages that ATPγS-driven backsteps originate pre-hydrolysis (Fig. 6, pink pathway), much earlier in the cycle than backslips (Fig. 6, orange pathway). In our model, ATPγS-driven backslips occur by essentially the same mechanism that they occur in ATP. Both forward steps and backslips need to wait for the initial ATPYS bound state to isomerise into hydrolytic competence, which adds a ~300 ms waiting time that dominates the overall dwell time at low loads. Summing the backstep and backslip dwell time populations further erodes the difference between the average forward and backward dwell times. Any residual differences between forward and backward average dwell times in ATPγS will tend to disappear into the noise.

## How does 1 mM ATPγS generate proportionally more backsteps than 1 μM ATPγS?

Our core hypothesis is that nucleotide binding to the ATP-waiting state populates the AI state and triggers a diffusive search by the tethered head for a binding site, but because the AI state is incompetent to dock the NL, this diffusive search is unsteered. As a result, diffusion-to-capture with coupled ADP release typically results in a backstep (Fig. 6, pink pathway). We propose that by hyper-enriching the AI state, ATPγS promotes flux along a side-branch of the scheme (pink) that leads to backstepping under load.

Figure 6b–d show illustrative backstepping sequences in ATPγS. The sequence in Fig. 6b and Supplementary Movie 1 produces a single backstep, whilst that in Fig. 6c and Supplementary Movie 2 produces 2 consecutive backsteps. In both cases, the binding of ATPγS to the MT-bound head of a 1HB dimer sanctions unsteered diffusion-to-capture of the partner head to its nearest available rearwards site, where it binds and undergoes MT-activated ADP release. In 1 μM ATPγS, (Fig. 6b), the newly-apo rear head will usually remain in its apo state whilst the forward head accesses its hydrolytic conformation (potentially via partial NL docking, see below), completes hydrolysis and thioPi release and detaches in its TRAPPED ADP conformation. Detachment of the lead head transfers the load to the apo head, completing a single 8 nm backstep and re-establishing the 1HB ATP-waiting state. The amplitude of this event is restricted to ~8 nm by the length of the NL tether. In this scenario, reversion to the 1HB Await-ATP state depends on the head that has just released ADP remaining in its apo state, and on the partner head retaining its newly-generated ADP (Fig. 6, pink pathway), allowing it to detach from the MT. These requirements will be met in 1 μM ATPγS, because the probability of ATPγS binding on the timescale of interest will be low (Fig. 6b; Supplementary Movie 1). Consistent with this, in 1 μM ATPγS we observe both an exponential load-dependence of dwell times (Fig. 3f) and an exponential load-dependence of the F/B ratio (Fig. 2h), resembling those seen in 1 μM ATP (Fig. 2f). This indicates that forward stepping in 1 μM ATPγS is routinely successful and suggests that following a backstep in 1 μM ATPγS the motor routinely reverts to its 1 HB waiting state (Fig. 6b). By contrast in 1 mM ATPγS, nucleotide binding is

saturated and rapid reversion to the 1HB Await-ATP state is correspondingly less likely (Fig. 6c; Supplementary Movie 2), as indicated by the reduced F/B ratio (Fig. 2g) and the much-reduced load-dependence of dwell times (Fig. 3d, e). Both effects reflect an excess of backsteps at substall loads. The proportion of these is almost load-independent, and amplitudes are almost invariably 8 nm (Fig. 4f, h), consistent with Fig. 6b, c and Supplementary Fig. 8.

Multiple other scenarios are possible, for example, side-stepping under load to an adjacent protofilament may occur[14,21]. It is possible that following ADP release from the newly-arrived rear head, the forward head could release its ADP, rather than release from the MT - backwards load is known to increase attached lifetimes in ADP and favour ADP release[29]. In this case, subsequent ATPγS binding to the forward head would initiate a futile cycle of ATPγS turnover, with the motor continuing to dwell in a 2HB state, giving rise to unusually long dwell times. Relatedly, rebinding of ADP, as explored in our ADP supplementation experiments (Fig. 5), will tend to trigger backwards translocations. The binding of ADP to the 1HB ATP-waiting state will tend to trigger a backslip or detachment (Fig. 6d; Supplementary Movie 3). Binding of ADP to the apo forward head of a 2HB wait state would, by contrast, generate extra 8 nm mechanical backsteps (Fig. 6e).

### Previous work on stepping in ATPγS

In earlier work on kinesin stepping in ATPγS, it was shown that at zero load, ATPγS-driven kinesin stepping remains processive and plus-end biased, but additional sidesteps and backsteps are detected, compared to equivalent measurements done in ATP[21]. It was suggested, because rates of half-site release driven by AMPPNP and ATPγS are slower than the overall rate of MT-activated ATP turnover, that stepping typically waits for hydrolysis[30,31]. In our proposed scheme for stepping under hindering load, forward stepping and hydrolysis are coupled, and the probability of backsteps is increased, in agreement with these earlier observations. Our interpretation differs from this previous work on stepping at zero load only in that we envisage that under load, neck linker docking promotes hydrolysis, rather than the other way around. We have no information on sidestepping.

### Partial NL docking can support a basal ATP hydrolysis rate

A subtle but crucially important feature of NL docking is that docking even of only the first residue of the NL can substantially accelerate hydrolysis[11,18]. In our scheme, hydrolysis can occur without full NL docking, for example, under superstall loads, at a default slow rate that feeds the backslip (Fig. 6a, orange) pathway and/or no-step (Fig. 6a, yellow) pathways. This rate corresponds to the slow load-independent backslipping rate observed under superstall load, which is about 2 s⁻¹. This slow default rate might reflect transient docking of the initial segment of the NL, as shown (Fig. 6a). In our working model, slow, weakly load-dependent flux along the yellow and orange pathways competes with strongly load-dependent forward stepping, defining a time window for forward stepping under load.

### Proposed role for the AI state in the biasing mechanism

We propose that in ATP, populating the AI state enables the kinesin motor to pause hydrolysis and wait for NL-steered leading head attachment to succeed. We envisage that backstepping from the AI state is a relatively slow 'leakage' pathway. If backstepping from the AI state occurred readily, then pausing in the AI state would entirely defeat the biasing mechanism. 1 mM ATPγS tends to defeat the biasing mechanism because the AI state is overpopulated and extra backsteps are generated at all substall loads. Nonetheless, even in 1 mM ATPγS, forward stepping is more probable than backstepping at all substall loads, and net directional bias is maintained. In summary, we propose that pausing in the AI state is key to the biasing mechanism, because it allows kinesin to wait in place without hydrolysing ATP, thereby

maximising steered diffusion-to-capture of the tethered head whilst maintaining tight coupling of forward stepping to ATP turnover. In ATP, we envisage that backstepping from the AI state occurs, but that backstepping from the AI state (pink pathway) is rare compared to backslips (orange pathway). By hyper-enriching the AI state, ATPγS enhances backstepping, revealing the pink pathway.

### Is a scheme without the AI state tenable?

Our working model (Fig. 6a) posits the existence of a pre-hydrolysis nucleotide-bound state, the AI state, that is activated for stepping but not for hydrolysis. Enrichment of this state in 1 mM ATPγS slows hydrolysis and undermines the biasing mechanism by generating extra 8 nm backsteps. We situate the AI state between the OPEN and CLOSED states, and on-pathway for both ATPγS and ATP. How secure are these features? Concerning the existence of the AI state, the crucial point is that the properties of the AI state are distinct from those of both the OPEN and CLOSED states. The OPEN state is an apo state that is not activated for stepping. The AI state has bound ATP or ATPγS and is activated for stepping but not for NL docking and hydrolysis. The CLOSED state has a bound nucleotide and is activated for NL docking and hydrolysis. If we attempt to collapse the properties of the AI state into the OPEN state, we end up triggering stepping without nucleotide binding, which is clearly wrong. If instead we attempt to collapse the properties of the AI state into the CLOSED state, we end up with a composite CLOSED state within which the motor proportionates between hydrolytically-competent and hydrolytically-incompetent substates according to the load. At one extreme (high load), a composite CLOSED state could have the same properties as the AI state (undocked NL, tethered head diffusing, hydrolytically incompetent). The effect of ATPγS would be to jam the composite state at this extremum. This scheme is equivalent to that which we propose, except that we identify this extremum as a separate state, based on its hydrolysis-incompetence. We think the working model we propose is the most parsimonius that can explain our data. In our proposed model, the AI state is critically important because it enables tight coupling of forward kinesin stepping to ATP turnover. By pausing in the AI state, the motor is enabled to wait for NL docking under load to steer the tethered head to its next on-axis binding site, facilitating isomerisation into the CLOSED state and the coupled hydrolysis of ATP.

We have argued that the mechanics of ATPγS-driven stepping under load reveal an Await-Isomerisation (AI) state of kinesin-1 that engenders slow backsteps. In the AI state, ATP is bound and stepping is potentiated, but NL docking and the coupled hydrolysis of ATP are not. In the scheme we propose, the AI state is transient at low load in ATP but enriched at high load in ATP, and at all loads in ATPγS. At the start of each mechanochemical cycle, nucleotide binding populates the AI state, allowing the motor to pause hydrolysis and wait for NL docking to steer the tethered head to its next on-axis microtubule binding site. NL docking potentiates hydrolysis on the MT-bound head. As the load increases, full NL docking and coupled diffusion-to-capture of the tethered head increasingly fail to occur during the forward-step time window. In this case, hydrolysis proceeds at a slower rate, allowing the motor to slip back, re-engage, and ultimately retry forward stepping. The ability to pause hydrolysis under load emerges as a central principle of the kinesin-1 mechanism, whose control logic maximises processive forward progress by adaptively blending tight-coupled forward steps with loose-coupled no-steps, backsteps and backslips.

## Methods

### Kinesin beads

Unmodified 560-nm polystyrene beads (Polysciences, Warrington, PA) were incubated with purified recombinant full-length *Drosophila* kinesin-1 (12) and diluted stepwise until approximately one-third of the beads displayed motility. Experiments were carried out in BRB80

buffer (80 mM K-PIPES, pH 7.0, 2 mM MgSO₄, 1 mM EGTA, 3 g/L glucose) supplemented with 1 mM ATP or ATPγS, a glucose-catalase oxygen scavenging system, and 10 mM Taxol. ATPγS (adenosine 5′-O-(3-thiotriphosphate)) was purchased from Roche (Cat. No. 11162306001).

## Microtubule preparation

Tubulin from porcine brain was diluted in 1× BRB80 buffer supplemented with 1 mM GTP to a final concentration of 20 μM. The solution was centrifuged at 90,000 rpm for 10 minutes, and the supernatant was incubated at 37 °C for 20 minutes to facilitate polymerization. Taxol was then added in a stepwise manner: first, 1.75 μL of 2 mM Taxol was added (final concentration ~75 μM), and the mixture was incubated at 37 °C for 10 minutes, followed by the addition of 1 μL of 2 mM Taxol (final concentration ~ 116 μM) and further incubation at 37 °C for 20 minutes. The microtubules were then maintained at room temperature for several hours before use in optical trap assays.

## Flow cells

The flow cell surface was passivated with 0.1 mg/mL casein (SuSoS AG, Dübendorf′, Switzerland). MTs were covalently attached to the coverslip surface using mild glutaraldehyde cross-linking to the 3-Aminopropyl)triethoxysilane (APTES)-silanized surface[32].

## Optical trapping

A custom-built optical trap[12] was used, equipped with a 3-W Nd:YAG 1064-nm laser (IE Optomech, Newnham, England). MTs were initially visualized by differential interference contrast microscopy, and beads were moved into position above the MTs by steering the trap. Imaging was then switched to amplitude contrast, and the image was projected onto the quadrant photodiode detector. Data were recorded at 20 kHz, and the moving average was filtered to 1 kHz during analysis.

Superstall experiments were performed as previously described by Carter and Cross[5]. Briefly, once kinesin reached a 3.5-pN trigger point (2.5 pN when ATPγS was used), the motor was subjected to a predefined high-force load by moving the microtubule (via a piezo-electric stage) toward the plus end. A transient, software-based force-feedback system was employed, utilizing both on-axis and off-axis bead position data while accounting for trap stiffness, microtubule orientation, and polarity. Microtubule displacement was typically completed within 200 ms, after which the feedback system was deactivated, and the trap and stage positions were held fixed. Experiments were repeated at least three times. Single-molecule results were reproduced across ≥10 independent bead-motor complexes per condition.

## Data analysis

Michaelis - Menten fits were performed using the drc package (function drm() with MM.2()), and summary statistics and plotting used base R packages/functions. Data were analyzed in R using custom-written code (available on request). Automated step-detection was implemented using t-test analysis. In the t-test analysis, twelve data points before the suspected step and twelve after the step were compared by t-test. Dwell times were defined as the waiting time between two consecutive steps. To ensure the accuracy of dwell time measurements, dwells in each force bin were manually verified. This verification involved visually inspecting the data to identify any missed steps between consecutive dwell times. In cases where a missed step was detected, the corresponding dwell was removed, as it represented a false dwell caused by the missed step. This manual inspection helped eliminate any false dwell times, ensuring the reliability of the data. Only steps above 2 pN could reliably be detected, and only these were processed. For forward step/backstep ratio measurements, the force range 3–8 pN was analyzed to ensure a sufficient number of both forward steps and backsteps. Each force bin includes data at the force shown ± 0.5 pN.

## Dwell time analysis and model fitting for Fig. 4

The plot represents the dwell time distribution for kinesin motor forward or backward steps within the specified force range (pN), displayed as a normalized 1-CDF (%). Dwell time is displayed on the x-axis using a log scale, while the y-axis shows the frequency of each dwell time occurrence, also on a log scale. For each force bin, dwell times were sorted, and the y-value at each dwell time was calculated as 1 minus the cumulative distribution function (1−CDF), normalized to 100% to allow comparison across conditions. To analyse the dwell time distribution, the normalized 1-CDF data were fitted to a single exponential decay model of the form $y = A\,e^{(-\lambda * x)}$ where $A$ is the amplitude and $\lambda$ is the decay constant. The mean dwell time $\tau$ was calculated as the inverse of the decay constant $1/\lambda$ and reported in seconds for each force bin. The obtained mean dwell time was then plotted against load, and the data were fitted using least-squares regression to the equation $\log(\tau) = k * load + b$.

## Reporting summary

Further information on research design is available in the Nature Portfolio Reporting Summary linked to this article.

## Data availability

Source data are provided with this paper.

## Code availability

Custom R codes are available from the corresponding author upon request.

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

## Acknowledgements
This work was funded by a Wellcome Investigator Award to R.A.C., grant number 220387/Z/20/Z.

## Author contributions
V.K. performed all the experiments and analyzed all the data. N.J.C. designed and built the optical trap. V.K., A.T., N.J.C., J.E.M and R.A.C. collaborated to design and interpret experiments and write the manuscript.

## Competing interests
The authors declare no competing interests.
