## [Transparent Peer Review file · Nature Communications]

ATP γ S substantially defeats the biasing mechanism for kinesin steps

Corresponding Author: Professor Robert Cross

Version 0:

Reviewer comments:

Reviewer #1

(Remarks to the Author)

In the manuscript titled "ATP γ S unbiases kinesin," the authors propose a novel intermediate state in the kinesin-1 mechanochemical cycle, the Await-Isomerisation (AI) state. The AI state is characterized as a pre-hydrolysis, one-head-bound conformation where the nucleotide is bound and the diffusional search of the tethered head is potentiated, but the catalytic site remains closed and incompetent for hydrolysis. To probe this state, the study employs ATP γ S to artificially overpopulate the AI state and reveal its kinetic and mechanical properties. The experimental approach employed is robust, and the result reasonably supports the proposed model. This manuscript is interesting and clearly written. I recommend this paper for publication in Nature Communications after considering the minor issue.

Comments:

1. Figure 1(d)-(f) has green solid lines, which seem to represent the various steps or slips taken by kinesin motors during their movement under the load. The authors should provide a detailed explanation of them.
2. In Figure 4(a)-(d), the y-axis is labeled as $1/\lambda$. While this is explained in the "MATERIALS AND METHODS" section, it would be helpful to include a brief definition in the main text as well. This will make readers easily understand the meaning of the graphs.

Reviewer #2

(Remarks to the Author)

The authors probe the kinesin chemomechanical cycle by analyzing the stepping patterns in ATP γ S under optical tweezer loads. This work builds on their previous analysis of kinesin stepping in ATP, where they identified a slip state at stall. Here, the stepping cycle is slowed down by ATP γ S and more frequent backsteps/slips are seen at saturating nucleotide but not at limiting nucleotide. From these measurements, they propose a modified kinesin hydrolysis cycle. This modified cycle can explain the Hackney ATP γ S/AMPPNP half-site data and gold tracking data in ATP γ S.

The strengths of the manuscript are the solid experimental data. The results section is written tersely and clearly. However, the interpretation is quite difficult to understand, even after multiple readings. I think that could be helped by a more informative Figure 6, perhaps by drawing out individual sequences that are common or preferred under different nucleotide conditions. The authors might consider writing the paper more as a combined results/discussion to give interpretation along the way. As it is now, the discussion is very dense and intense to get through.

My opinion is that the hydrolysis cycle could be simplified by postulating that a forward step requires ATP hydrolysis and describing the ATP γ S effect as slowed hydrolysis rather than slowed pre-closed to closed isomerization. This is essentially removing state 3 in the main pathway. The authors should justify why they need to add a new state. It would be nice if the cycle could be simplified this way.

The authors make clear predictions about sidesteps in ATP γ S. It would seem that if they occur then they should be evident in their optical tweezer traces. If they didn't, then this is very relevant and if they are, it would support the model strongly.

Specific comments:

The authors don't say it explicitly, but I believe they are implicitly saying that the 8 and 16 nm backslips seen in ATP are fundamentally different than the 8 nm tethered backsteps seen in 1 mM ATPgS. It is a point that easy to miss.

I may be mistaken, but it isn't clear to me why in ATPgS detachments should preferentially occur after backsteps – the tethered backstep pathway resolves to the main pathway in Fig 6. Is it just because there are more backsteps? In the Guydosh and Block work 127-130, the obligate backstep was due to trapping of AMPPNP in the trailing head in the 2HB state. I don't see the relevance.

It is surprising and not explained why in Fig. 3c there is such a difference between the forward and backward stepping dwell times. The delay scales with the times (lines are parallel). This substantial delay isn't accounted for in the hydrolysis cycle presented.

I still can't quite understand why at 1 μ M ATPgS the load dependence of the for/back ratio (2d) and dwell times changes compared to 1 mM ATPgS.

Fig 1 – define green lines

Add ref in line 77

Line 90/91 – isn't this saying that ATPgS binding is load dependent?

Reviewer #3

(Remarks to the Author)

In this paper, Karnawat et al. measured single-molecule stepping motion of kinesin-1 under an optical trap load in the presence of ATPg(γ)S, a slowly-hydrolyzing ATP-analog, and compared this motility with that driven by ATP. They discovered unique features of ATPgS-driven motility; 1) the forward/backward step ratio decreases compared to ATP and remains nearly independent of load magnitude, 2) dwell times before forward and backward steps become similar, and 3) backward steps are predominantly 8-nm steps even with added ADP. Reducing ATPgS concentration to 1 μ M increased the forward/backward step ratio to levels similar to those with ATP. To reconcile these results, the authors proposed a revised model that includes an "await-isomerization (AI) state" preceding the transition to closed and neck linker docked conformation, which is stabilized by ATPgS and leads to off-pathway binding of the tethered head to the rear binding site on the adjacent protofilament. While the experimental results are interesting, their interpretation is incomprehensible. It relies on unsubstantiated hypotheses and presents disjointed reasoning, making the discussion difficult to follow. The proposed model fails to provide new insight into the basic principle for kinesin motility under load and instead potentially explains unusual cases, such as when kinesin overcomes an obstacle. Therefore, the significance of this work in the field is limited.

Their discussion entirely relies on the hypothesis of the AI state, which they define as a pre-closed conformation where nucleotide is bound and diffusional excursions of the tethered head are potentiated, but neck linker docking is suppressed. They claim that ATPgS and ATP under load retard the isomerization to hydrolysis-competent closed state, thereby accumulating the AI state. However, the authors provide neither experimental evidence nor logical justification for this hypothesis. It is more reasonable to propose that ATPgS stabilizes the closed conformation while preventing ATP hydrolysis, similar to the well-studied ATP analog, AMPPNP. X-ray crystallography and cryoEM structures of kinesin without neck linker constraint in complex with AMPPNP showed a closed conformation in which the nucleotide pocket is closed and the neck linker is fully docked. CryoEM images by Benoit et al. (2021) demonstrated that the AMPPNP-bound leading head, with its neck linker pulled backward, adopts an "open" conformation with an opened nucleotide pocket. However, this does not represent the AI state without neck linker constraint (rather it is an open state), as Niitani et al. (2025) demonstrated that the nucleotide affinity of the leading open head is significantly lower than that of the trailing closed state. The structural difference between AMPPNP and ATPgS is small compared to the size of kinesin's nucleotide pocket, making it implausible that kinesin with ATPgS adopts a distinct conformation. Therefore, it is crucial that the authors provide experimental evidence to validate this hypothesis (e.g., cryoEM images of the kinesin-microtubule complex bound to ATPgS).

To explain the observation that backsteps in the presence of ATPgS were almost 8 nm steps, they introduced what appears to be an implausible hypothesis termed "tethered backslip". In this hypothesis, the tethered head in AI state can bind to an 8-nm backward binding site on the neighboring protofilament, after which the microtubule-bound head detaches and is pulled backward. Such sideways stepping has never been observed for wild-type kinesin and is only observed in neck linker extended mutants, as the neck linker length cannot accommodate a two-head-bound state where heads bind to adjacent protofilaments. The authors cite the Mickolajczyk paper (PNAS 2015) regarding stepping motion in the presence of ATPgS, but their off-axis displacements are significantly larger (10-20 nm) than the spacing between adjacent protofilaments (~5 nm), and their gold probes are likely multivalent, causing unsynchronized binding and detachment of multiple kinesin molecules bound to the bead. The 8-nm backsteps have traditionally been explained as an off-pathway transition initiating from the two-head-bound state, not from a one-head-bound state as the authors proposed. The leading head hydrolyzes ATP and detaches from the microtubule before the trailing head, and the detached head binds to the rear binding site on the same protofilament 16-nm backward from the original site, causing an 8-nm backward displacement of the trapped bead. The detachment of the leading head in the two-head-bound state rarely occurs when the trailing head efficiently hydrolyzes

ATP. However, it might become significant when this state is prolonged, giving the leading head more opportunity to detach.

The authors appear not to take into account the duration of the two-head-bound state in interpreting the dwell times between steps. The observed dwell times before forward and backward steps (except slips) include both those in the two- and one-head-bound states. If ATPgS stabilizes the closed state but hydrolysis does not readily proceed, kinesin with ATPgS spends most of the time in the two-head-bound state (with the ATPgS-bound head trailing) even under high load. The dwell time in the one-head-bound state becomes longer when hindering load increases in the presence of ATP (~200 ms at 7 pN; Fig.3), but is still shorter than the turnover rate of ATPgS (~700 ms, estimated from the gliding speed) and the dwell time between steps with ATPgS at low load (~ 500 ms; Fig. 3). In the presence of 1 μ M ATPgS, nucleotide binding becomes slower (~250 ms if we assume the on-rate is similar to that of ATP (2-5 μ M \cdot 1s $^{-1}$)), and the molecule spends comparable time in both one- and two-head-bound states. As described above, the 8-nm backsteps could originate from the two-head-bound state. Therefore, the dwell time ratio between the two-head-bound state (which originates 8-nm processive backward step) and the one-head bound state (which originates 8-nm forward step and backward slip) accounts for the experimental results without requiring any new assumptions.

Minor comments:

- 1) The title is misleading and incorrect, since kinesin moves unidirectionally under hindering load up to 7 pN in the presence of ATPgS (Fig. 1).
- 2) The source of ATPgS should be specified in the method section. Are there any contaminations of side products or ADP?
- 3) Fig. 1d-g: the y-axes on the left side should display the displacement of the bead, since displacement is the direct measurement. The load calculation, which includes uncertainty due to errors in determining trap stiffness and deviations from linearity, should be moved to the right side of the y-axis.
- 4) Fig. 2: the color for > 12 nm backward steps appear to be brown, not magenta.
- 5) Fig. 2i-l: are the data for 2 and 3 pN (before reaching the trigger point) the same as those in Fig. 2a and c? The authors should distinguish between data with and without rapid movement of the stage.
- 6) Fig. 2j,l: typical traces of 1 mM ATPgS under superstall forces should be shown. Are there successive 8 nm backward steps (Fig.1e appears to shows successive (processive) backsteps)?
- 7) Fig. 4a-d: the populations of longer dwell times deviated from the fit. Did the authors attempt to fit with a double exponential?
- 8) Fig. 4a-d: what is the definition of λ (τ might be common)? This has been written only in the method section but should be clarified in the text or the figure legend.
- 9) Fig. 4e-h: what is the concentration of the nucleotides?
- 10) Fig. 4e-h: y-axis label is missing.
- 11) Figs.3,4: the authors displayed only two examples of the histograms in Fig. 4. All histogram data with fit used to determine the mean dwell times in Figs. 3 and 4 should be provided as supplementary data or at a repository site to evaluate the accuracy of the mean dwell times.

Reviewer #4

(Remarks to the Author)

Version 1:

Reviewer comments:

Reviewer #1

(Remarks to the Author)

The authors have made significant revisions to the manuscript, leading to considerable improvements. This work presents intriguing results in the field and is suitable for publication in Nature Communications.

Minor Comment:

The forward-to-backward probability ratios shown in Figure 2 seem to be derived from the trajectories of beads trapped in optical tweezers, such as those presented in Figures 1d-g. It would be helpful to describe this explicitly.

Reviewer #2

(Remarks to the Author)

Reviewer #3

(Remarks to the Author)

The authors have responded carefully to the concerns I raised, but my greatest concern remains unanswered. For an 8 nm backstep to occur, two highly improbable transitions must happen consecutively: 1) the tethered head binds to the rear-binding site, not the forward-binding site, in the one-head-bound state, and 2) the front head detaches from the microtubule before the rear head in the two-head-bound state. The probability of the first transition increases as the hindering load increases, since the tethered head is moved backward by the load. However, in the presence of ATP, even when this first transition occurs, the second transition is likely prohibited, that is, the rear head detaches before the front head, causing futile ATP hydrolysis and a 0-nm step that cannot be detected by optical trapping assays (the pink pathway in Figure 6 is incorrect, as it branches and returns to the main pathway in most cases before a backstep occurs). This occurs due to gating mechanisms in the two-head-bound state, which stabilizes the front head in an open conformation via the backward-pulling neck linker and the rear head in a closed conformation via the forward-stretched neck linker, as has been demonstrated recently. Guydosh and Block (PNAS 2006) also demonstrated that AMPPNP or ADP/BeFx binds tightly to the rear head but can be released after the head moves to the front position.

Therefore, for the second transition to occur, there must be a mechanism to overcome the front and rear head gating mechanisms. However, the authors focus on explaining the first transition and do not provide a convincing explanation for the second. Starting from the AI state in Figure 6, when the tethered head binds to the rear binding site, the front head would more likely release bound ATPgS rather than hydrolyzing it, as has been demonstrated by Guydosh and Block. Then, ATPgS binds to the closed rear head, and its hydrolysis proceeds very slowly. But since the closed conformation of the rear head is more compatible with the hydrolysis reaction, the rear head would detach before the front head, causing futile ATPgS hydrolysis and a 0-nm step. The authors proposed that partial docking of the neck linker in the front head allows ATPgS hydrolysis. However, the microtubule-detachment rate would decrease substantially when the neck linker is pulled backward compared to when it is pulled forward as demonstrated by Niitani et al (2025).

I suspect contaminating ADP is essential to break the front head gating mechanism. The ATPgS used by the authors (Roche 11162306001) contains a non-negligible amount of ADP. I examined the Certificate of Analysis for five lots on their website (sigmaaldrich.com) and found that ADP ranges from 2.6-7.5% (4.3% on average). Under 1 mM ATPgS conditions, the contaminating ADP concentration is approximately 4 μ M, corresponding to an ADP binding rate of about 15 s⁻¹ (once every 60 ms), which is faster than the ATPgS turnover rate. This ADP-induced detachment of the front head can explain the 8-nm backstep observed with 1 mM ATPgS. In contrast, under 1 μ M ATPgS condition, the ADP binding rate decreases to approximately 0.15 s⁻¹, much slower than the ATPgS turnover rate. Under this condition, the rear head is more likely to detach before to the front head. Therefore, which head detaches from the microtubule first can be explained by whether detachment due to ATP or ATPgS hydrolysis or detachment promoted by contaminating and/or added ADP binding occurs first. Combining these considerations for the second transition with the load dependence of the first transition provides explanations for all the experimental results.

I believe the authors should address these two concerns, a reasonable explanation for breaking the front head gating and the potential role of contaminating ADP in breaking the front head gating, before the manuscript can be accepted.

Reviewer #4

(Remarks to the Author)

Version 2:

Reviewer comments:

Reviewer #3

(Remarks to the Author)

The authors have satisfactorily addressed my concerns, and I support the publication of this work.

REVIEWER COMMENTS

Reviewer #1 (Remarks to the Author):

In the manuscript titled "ATP γ S unbiases kinesin," the authors propose a novel intermediate state in the kinesin-1 mechanochemical cycle, the Await-Isomerisation (AI) state. The AI state is characterized as a pre-hydrolysis, one-head-bound conformation where the nucleotide is bound and the diffusional search of the tethered head is potentiated, but the catalytic site remains closed and incompetent for hydrolysis. To probe this state, the study employs ATP γ S to artificially overpopulate the AI state and reveal its kinetic and mechanical properties. The experimental approach employed is robust, and the result reasonably supports the proposed model. This manuscript is interesting and clearly written. I recommend this paper for publication in Nature Communications after considering the minor issue. Comments:

1. Figure 1(d)-(f) has green solid lines, which seem to represent the various steps or slips taken by kinesin motors during their movement under the load. The authors should provide a detailed explanation of them.
2. In Figure 4(a)-(d), the y-axis is labeled as $1/\lambda$. While this is explained in the "MATERIALS AND METHODS" section, it would be helpful to include a brief definition in the main text as well. This will make readers easily understand the meaning of the graphs.

Thank you for these supportive comments. We have addressed the two minor issues raised as follows:

1. We now provide an explanation within the relevant figure legends of the step-finding procedure used to generate the green overlays. We previously described this only in the Methods section, without pointing to the green overlay.

Fig. 1 legend line 463: "Steps marked in green are detected based on their t-score cutoff (see Methods)."

2. At the reviewer's suggestion, we now explain the meaning of $1/\lambda$ within the main text ($1/\lambda = \tau$).

Results line 137: "*The characteristic dwell time τ , the inverse of the fitted decay constant, is obtained.*"

Methods line 448: "*The mean dwell time τ was calculated as the inverse of the decay constant ($1/\lambda$) and reported in seconds for each force bin.*"

Fig. 4 legend line 495/496: "*mean dwell time ($\tau = 1/\lambda$) vs. load*"

Reviewer #2 (Remarks to the Author):

The authors probe the kinesin chemomechanical cycle by analyzing the stepping patterns in ATP γ S under optical tweezer loads. This work builds on their previous analysis of kinesin stepping in ATP, where they identified a slip state at stall. Here, the stepping cycle is slowed down by ATP γ S and more frequent backsteps/slips are seen at saturating nucleotide but not at limiting nucleotide. From these measurements, they propose a modified kinesin hydrolysis cycle. This modified cycle can explain the Hackney ATP γ S /AMPPNP half-site data and gold tracking data in ATP γ S .

The strengths of the manuscript are the solid experimental data. The results section is written tersely and clearly.

Thank you for these supportive comments.

However, the interpretation is quite difficult to understand, even after multiple readings. I think that could be helped by a more informative Figure 6, perhaps by drawing out individual sequences that are common or preferred under different nucleotide conditions. The authors might consider writing the

paper more as a combined results/discussion to give interpretation along the way. As it is now, the discussion is very dense and intense to get through.

We appreciate these frank comments and have reworked Fig. 6 and our Discussion to parse alternative possible stepping scenarios, as the reviewer urges. Also as suggested, we have combined the Results and Discussion sections, aiming to explicitly address the points raised by the reviewers, and others, under subheadings.

My opinion is that the hydrolysis cycle could be simplified by postulating that a forward step requires ATP hydrolysis and describing the ATP γ S effect as slowed hydrolysis rather than slowed pre-closed to closed isomerization. This is essentially removing state 3 in the main pathway. The authors should justify why they need to add a new state. It would be nice if the cycle could be simplified this way.

Our working model, with its extra AI state, is prompted by our evidence that ATP γ S generates extra 8 nm backsteps, compared to ATP. We infer that this is because for ATP γ S, the isomerisation into hydrolysis-competence is retarded and that the pre-isomerisation state, with ATP γ S bound but unhydrolyzed, generates unsteered steps. The steps are unsteered because pre-isomerisation, the NL cannot dock, because its binding site does not exist. ATP γ S backsteps are 8 nm only, and not a spectrum of 8, 16, 24 nm. as seen in ATP, because they are steps and not slips. Our data are obtained under load. This allows us to detect backstepping from the ATP γ S-bound state under conditions where NL docking and forward stepping are inhibited by the load. The probability of ATP γ S-driven backstepping from the AI state is almost load-independent (Fig. 2b), and the dwell times for backsteps at saturating ATP γ S are almost load-independent (Fig. 4b). We take both these observations as evidence that ATP γ S-driven backstepping occurs from a pre-hydrolysis ATP or ATP γ S state that cannot dock the NL.

We have inserted a new paragraph into the Results and Discussion in which we explicitly consider whether a scheme without an AI state can work:

Line 341: "Is a scheme without the AI state tenable?"

Our working model (Fig. 6a) posits the existence of a pre-hydrolysis nucleotide-bound state, the AI state, that is activated for stepping but not for hydrolysis. Enrichment of this state in 1 mM ATP γ S slows hydrolysis and undermines the biasing mechanism by generating extra 8 nm backsteps. We situate the AI state between the OPEN and CLOSED states, and on-pathway for both ATP γ S and ATP. How secure are these features? Concerning the existence of the AI state, the crucial point is that the properties of the AI state are distinct from those of both the OPEN and CLOSED states. The OPEN state is an apo state that is not activated for stepping. The AI state has bound nucleotide and is activated for stepping but not for NL docking and hydrolysis. The CLOSED state has bound ATP or ATP γ S and is activated for NL docking and hydrolysis. If we attempt to collapse the properties of the AI state into the OPEN state, we end up triggering stepping without nucleotide binding, which is clearly wrong. If instead we attempt to collapse the properties of the AI state into the CLOSED state, we end up with a composite CLOSED state within which the motor proportionates between hydrolytically-competent and hydrolytically-incompetent conformations according to the load. At one extreme (high load), a composite CLOSED state could have the same properties as the AI state (undocked NL, tethered head diffusing, hydrolytically incompetent). The effect of ATP γ S would be to jam the composite state at this extremum. This scheme is equivalent to that which we propose, except that we identify this extremum as a separate state, based on its hydrolysis-incompetence. We think the working model we propose is the most parsimonious that can explain our data. In our proposed model, the AI state is critically important because it enables tight coupling of forward kinesin stepping to ATP turnover. By pausing in the AI state, the motor is enabled to wait for NL docking under load to steer the tethered head to its next on-axis binding site, facilitating isomerisation into the CLOSED state and the coupled hydrolysis of ATP. "

Note that during ATP-driven stepping at zero load, we expect the AI state to exist only very transiently. Under these conditions the simpler scheme proposed by the reviewer is fully appropriate. We expect the AI state to play a significant mechanistic role only when NL docking is opposed by an external load.

Under load, the existence of the hydrolysis-incompetent AI state allows the motor to pause hydrolysis and wait for tethered diffusion to carry the leading head to its next site, at which point coupled NL docking stabilises the CLOSED state, triggering hydrolysis.

The authors make clear predictions about sidesteps in ATP γ S . It would seem that if they occur then they should be evident in their optical tweezer traces. If they didn't, then this is very relevant and if they are, it would support the model strongly.

We infer sidestepping, based on previous evidence for erratic stepping in ATP γ S at zero external load (ref 21). We do not claim to have detected sidestepping. Sidestepping is not central to our case, whereas backstepping is. We have amended the main text to remove almost all reference to sidestepping, other than mentioning it as a possibility.

Line 295: "Multiple other scenarios are possible, for example side-stepping under load to an adjacent protofilament may occur."

Line 315: "We have no information on sidestepping".

Specific comments:

The authors don't say it explicitly, but I believe they are implicitly saying that the 8 and 16 nm backslips seen in ATP are fundamentally different than the 8 nm tethered backsteps seen in 1 mM ATP γ S . It is a point that easy to miss.

We intended to be explicit about this point and we have revised our text and figures to reinforce:

Para beginning line 160: "Added ADP provokes backslips in ATP but backsteps in ATP γ S"

Legend to Figure 6: "... In b, ATP γ S (S) binding triggers a single 8 nm backstep with reversion to the AA state. This behaviour depends on the rear head remaining in its apo state whilst the lead head is recovered and so will dominate at 1 μ M ATP γ S. In c, ATP γ S binds earlier, causing a second 8 nm backstep. This behaviour will dominate at 1 mM ATP γ S. d is an 8 nm backslip, caused by ADP rebinding during ATP-driven stepping. e is an 8 nm backstep, caused by ADP rebinding in ATP γ S."

I may be mistaken, but it isn't clear to me why in ATP γ S detachments should preferentially occur after backsteps – the tethered backstep pathway resolves to the main pathway in Fig 6. Is it just because there are more backsteps?

The reason that a backstep routinely precedes a detachment in ATP γ S but not ATP is that in ATP γ S but not in ATP, the motor needs to revert from a 2HB state to a 1HB state (causing a backstep) before it can detach. In ATP the motor is already in a 1HB state and can detach directly. This is now explicitly illustrated using frame-by-frame diagrams (Fig. 6b,c,d) and animations (Supp. Movies 1-3).

In the Guydosh and Block work, the obligate backstep was due to trapping of AMPPNP in the trailing head in the 2HB state. I don't see the relevance.

The Guydosh and Block is relevant only to the extent that nucleotide release is controlling a backstep. With hindsight referring to Guydosh & Block at this point is confusing and we have now removed it (we already reference the Guydosh & Block work in our Introduction). To explain backstepping we now provide a frame-by-frame diagram detailing how ATP γ S gives rise to a 2HB waiting state, which must revert to a 1 HB Await ATP state as a precondition for detachment. (Fig. 6b,c). We predict that added ADP will promote this reversion (Fig 6d,e). For added clarity we also provide animations (Supp. Movies 1, 2, 3).

It is surprising and not explained why in Fig. 3c there is such a difference between the forward and backward stepping dwell times. The delay scales with the times (lines are parallel). This substantial delay isn't accounted for in the hydrolysis cycle presented.

The time gap arises as the difference between the average dwell times for forward stepping and backslipping (New Fig. 3e; New Fig. 4c). In previous work, we found that this difference is increased by certain MT lattices. We speculated that this is because Pi release is faster on some lattices than others, but we have no direct evidence. In our model, Pi release closes the forward step time window. Before Pi release, only forward steps are seen, whilst after Pi release, only backslips are seen. As a result, backslips have longer average dwell than forward steps because backslip dwells include time spent before Pi release trying and failing to step forward.

We have inserted a short section explaining this:

Line 232: " *Why are backwards dwells longer on average than forwards dwells?*

In our working model (Fig. 6), isomerisation of the AI state into the CLOSED state opens a 'try-forward' time window during which the motor tries to dock the NL under load. Forward steps (blue pathway) occur when it succeeds, backslips (orange pathway) when it does not. The try-forward time window is closed by Pi release. Average backstep dwells are longer than average forward step dwells because the dwell times for backslips contain time spent before Pi release attempting but failing to step forward. "

I still can't quite understand why at 1 μM ATP γS the load dependence of the for/back ratio (2d) and dwell times changes compared to 1 mM ATP γS .

We are very grateful for this comment because this point is key and we failed in the original ms to make it sufficiently clear. In 1 mM ATP γS , we see runs of 8 nm mechanical backsteps, we think because following ADP release, ATP γS tends to rebind quickly and trigger further unsteered backstep(s) (Fig. 6b). By contrast in 1 μM ATP γS , ATP γS binding is much slower, which increases the probability the motor will revert to its 1HB wait state, from which forward stepping is possible (Fig. 6c). This explains why backward steps are more probable in 1 mM ATP γS than in 1 μM ATP γS .

Fig 1 – define green line.

Done. This point was picked up also by R1.

Fig 1 legend line 463: "Steps marked in green are detected based on their t-score cutoff (see Methods)."

Add ref in line 77.

Done

Line 90/91 – isn't this saying that ATP γS binding is load dependent?

In our model, the collision complex with nucleotide forms load-independently. The subsequent NL docking step, which stabilises the hydrolysis-competent CLOSED state, is load-dependent. At high load, NL docking is hindered, slowing entry into the CLOSED state and extending dwell times; at low load, NL docking occurs readily, accelerating hydrolysis and forward stepping. Thus, the observed load dependence of dwell times reflects the NL docking step after nucleotide binding, rather than a direct effect of load on the binding step itself.

More exactly, whilst the binding of ATP or ATP γS is load-independent, the competing reaction, the unbinding of nucleotide from the AI state, will be indirectly load-dependent, because it will depend on the occupancy of the AI state, and exit from the AI state is coupled to load-dependent NL docking. We

suspect that the off rate for nucleotide from the AI state may be low, leading to little or no overall load-dependence. Investigating this point will require quantitative simulation. We hope to do this shortly.

Reviewer #3 (Remarks to the Author):

In this paper, Karnawat et al. measured single-molecule stepping motion of kinesin-1 under an optical trap load in the presence of ATP γ S, a slowly-hydrolyzing ATP-analog, and compared this motility with that driven by ATP. They discovered unique features of ATP γ S-driven motility; 1) the forward/backward step ratio decreases compared to ATP and remains nearly independent of load magnitude, 2) dwell times before forward and backward steps become similar, and 3) backward steps are predominantly 8-nm steps even with added ADP. Reducing ATP γ S concentration to 1 μ M increased the forward/backward step ratio to levels similar to those with ATP. To reconcile these results, the authors proposed a revised model that includes an "await-isomerization (AI) state" preceding the transition to closed and neck linker docked conformation, which is stabilized by ATP γ S and leads to off-pathway binding of the tethered head to the rear binding site on the adjacent protofilament.

While the experimental results are interesting, their interpretation is incomprehensible. It relies on unsubstantiated hypotheses and presents disjointed reasoning, making the discussion difficult to follow. The proposed model fails to provide new insight into the basic principle for kinesin motility under load and instead potentially explains unusual cases, such as when kinesin overcomes an obstacle. Therefore, the significance of this work in the field is limited.

We are glad the reviewer finds our results interesting and we are happy to have the chance to address their criticisms. We believe that our arguments are sound and that our data speak not to a special case, but to the kinesin mechanism in general.

Their discussion entirely relies on the hypothesis of the AI state, which they define as a pre-closed conformation where nucleotide is bound and diffusional excursions of the tethered head are potentiated, but neck linker docking is suppressed. They claim that ATP γ S and ATP under load retard the isomerization to hydrolysis-competent closed state, thereby accumulating the AI state. However, the authors provide neither experimental evidence nor logical justification for this hypothesis. It is more reasonable to propose that ATP γ S stabilizes the closed conformation while preventing ATP hydrolysis, similar to the well-studied ATP analog, AMPPNP.

Indeed both AMPPNP and ATP γ S inhibit hydrolysis, but in different ways. In AMPPNP, the active site is free to adopt its hydrolytically-competent CLOSED conformation, but it cannot hydrolyse the beta-gamma linkage. By contrast in ATP γ S, the extra bulk of the gamma thiophosphate hinders the transition into the hydrolytically-competent CLOSED conformational state of the active site. Once this kinetic barrier is overcome, hydrolysis proceeds. In response to the reviewer's comment, we now explicitly review the differences between ATP γ S and AMPPNP:

Line 62: "There is currently no structure for a kinesin-ATP γ S complex, which is of course only transiently stable, but there is a structure for myosin-ATP γ S, obtained using a small molecule inhibitor that blocks hydrolysis. In this myosin structure, the gamma thiophosphoryl of ATP γ S is rotated to allow it to project away from the catalytic centre. It is possible that a similar rotation is required in kinesin, and that the time required slows hydrolysis. The explicitly steric action of ATP γ S in slowing entry to the CLOSED state contrasts with the action of another widely-adopted nonhydrolyzable analogue, AMPPNP²³. In AMPPNP, kinesin readily adopts a CLOSED state, but hydrolysis is inhibited because in AMPPNP the hydrolysable P-O-P linking the beta and gamma phosphates of ATP is replaced with a P-N-P linkage. The exact mechanism by which ATP γ S slows kinesin-driven hydrolysis is not yet fully clear, but it is clear nonetheless that ATP γ S is a very useful tool."

X-ray crystallography and cryoEM structures of kinesin without neck linker constraint in complex with AMPPNP showed a closed conformation in which the nucleotide pocket is closed and the neck linker is fully docked. CryoEM images by Benoit et al. (2021) demonstrated that the AMPPNP-bound leading head, with its neck linker pulled backward, adopts an "open" conformation with an opened nucleotide pocket. However, this does not represent the AI state without neck linker constraint (rather it is an open state), as Niitani et al. (2025) demonstrated that the nucleotide affinity of the leading open head is significantly lower than that of the trailing closed state.

We suspect the AMPPNP-bound lead head in the Benoit et al cryoEM work (on kinesin-3) may resemble our proposed AI state quite closely. We do not think the demonstration that this state has a lower nucleotide affinity than the closed state gainsays this possibility.

The cryoEM images of Benoit et al. indeed shows that the AMPPNP-bound leading of kinesin-3, with its neck linker pulled backward, adopts an open conformation. Although this was described as an "open" conformation, we suggest that this conformation may correspond closely to the AI state in our scheme, in which nucleotide is bound but hydrolysis and NL docking are delayed. The observation by Niitani et al. that this leading head conformation has lower nucleotide affinity than the trailing closed state is fully consistent with our interpretation, as the AI state is expected to equilibrate with the CLOSED state and thus would be expected to differ in nucleotide affinity. Thus, rather than contradicting our proposal, these structural observations provide support for the existence of a nucleotide-bound pre-hydrolysis state that matches the functional characteristics we assign to AI state.

The structural difference between AMPPNP and ATP γ S is small compared to the size of kinesin's nucleotide pocket, making it implausible that kinesin with ATP γ S adopts a distinct conformation.

We disagree here, and as noted above, now explicitly consider this point.

Therefore, it is crucial that the authors provide experimental evidence to validate this hypothesis (e.g., cryoEM images of the kinesin-microtubule complex bound to ATP γ S).

A cryoEM structure with ATP γ S bound would be helpful, but unfeasible with the WT motor because ATP γ S is hydrolysed. This is the reason no one has yet produced an ATP γ S kinesin structure. The myosin structure mentioned above was possible only because a small molecule effector was available that allosterically blocks hydrolysis.

To explain the observation that backsteps in the presence of ATP γ S were almost 8 nm steps, they introduced what appears to be an implausible hypothesis termed "tethered backslip". In this hypothesis, the tethered head in AI state can bind to an 8-nm backward binding site on the neighboring protofilament, after which the microtubule-bound head detaches and is pulled backward. Such sideways stepping has never been observed for wild-type kinesin and is only observed in neck linker extended mutants, as the neck linker length cannot accommodate a two-head-bound state where heads bind to adjacent protofilaments.

The term 'tethered backslip' was not meant to imply sidestepping and clearly, with hindsight, was unhelpful. We have removed it, focussing instead on specific frame-by-frame scenarios as urged by R2. The pink (originally magenta) pathway in Fig. 6 has been redrawn to make it on-axis, consistent with our view that most backsteps from the AI state are on-axis. We mention the possibility that sidesteps occur but we have no evidence for sidesteps and we now say so.

Line 315: "We have no information on sidestepping."

The authors cite the Mickolajczyk paper (PNAS 2015) regarding stepping motion in the presence of ATP γ S, but their off-axis displacements are significantly larger (10-20 nm) than the spacing between adjacent protofilaments (~5 nm), and their gold probes are likely multivalent, causing unsynchronized binding and detachment of multiple kinesin molecules bound to the bead.

We take it that the Mickolajczyk paper compared the same kinesin-beads in ATP versus ATP γ S. The evidence that stepping by the same beads is more erratic in ATP γ S than in ATP then seems clear, and this is all we wish to take away.

The 8-nm backsteps have traditionally been explained as an off-pathway transition initiating from the two-head-bound state, not from a one-head-bound state as the authors proposed. The leading head hydrolyzes ATP and detaches from the microtubule before the trailing head, and the detached head binds to the rear binding site on the same protofilament 16-nm backward from the original site, causing an 8-nm backward displacement of the trapped bead. The detachment of the leading head in the two-head-bound state rarely occurs when the trailing head efficiently hydrolyzes ATP. However, it might become significant when this state is prolonged, giving the leading head more opportunity to detach.

Our model (Fig. 6a) posits 2 classes of backwards displacements, backslips from a 1HB waiting state (orange pathway in Fig. 6a) and backsteps from the proposed AI state (pink pathway in Fig. 6a). Backstepping from the AI state is enhanced by ATP γ S and its mechanism closely resembles that which the reviewer describes.

The authors appear not to take into account the duration of the two-head-bound state in interpreting the dwell times between steps. The observed dwell times before forward and backward steps (except slips) include both those in the two- and one-head-bound states. If ATP γ S stabilizes the closed state but hydrolysis does not readily proceed, kinesin with ATP γ S spends most of the time in the two-head-bound state (with the ATP γ S-bound head trailing) even under high load.

Under load in the optical trap, we take it that the load is born overwhelmingly by the forward (plus-endwards) head of an attached dimer. Dwells correspond to the interval over which the load is born by this one head. Stepping swaps the load to the other head and starts a new dwell. Our method is 'blind' to the time required to detach the previously-loaded head, though clearly this is included in the measured dwell time.

The dwell time in the one-head-bound state becomes longer when hindering load increases in the presence of ATP (~200 ms at 7 pN; Fig. 3), but is still shorter than the turnover rate of ATP γ S (~700 ms, estimated from the gliding speed) and the dwell time between steps with ATP γ S at low load (~500 ms; Fig. 3). In the presence of 1 μ M ATP γ S, nucleotide binding becomes slower (~250 ms if we assume the on-rate is similar to that of ATP (2-5 μ M⁻¹s⁻¹)), and the molecule spends comparable time in both one- and two-head-bound states. As described above, the 8-nm backsteps could originate from the two-head-bound state. Therefore, the dwell time ratio between the two-head-bound state (which originates 8-nm processive backward step) and the one-head bound state (which originates 8-nm forward step and backward slip) accounts for the experimental results without requiring any new assumptions.

This interpretation has much in common with the scheme we are proposing. We not only agree that it is possible that backsteps in ATP γ S can originate from a 2HB state, we argue that they do. But we disagree that no new features (states) are required to account for our data. Partly this is semantic – the reviewer suggests by implication that a CLOSED state can exist for which hydrolysis is slow. Whereas we define the CLOSED state by its ability to catalyse rapid hydrolysis. If nucleotide is bound but hydrolysis does not proceed, we are by (our) definition not in the CLOSED state, and we need to postulate the AI state as a pre-hydrolysis intermediate that supports stepping but not steering (NL docking). Please also see our response to R2 – in our view a new state is required because otherwise mutually contradictory properties need to be attributed to the same state.

Minor comments:

1) The title is misleading and incorrect, since kinesin moves unidirectionally under hindering load up to 7 pN in the presence of ATP γ S (Fig. 1).

In hindsight, we agree. We did not mean to imply by “unbiases” that ATP γ S entirely defeats directional stepping. Saturating concentrations of ATP γ S substantially defeat directional stepping, because at all substall loads, there are many extra backsteps. We have adjusted the title to reflect this.

New title: “ATP γ S substantially defeats the biasing mechanism for kinesin steps”

2) The source of ATP γ S should be specified in the method section. Are there any contaminations of side products or ADP?

We now state the source of ATP γ S.

Line 398/399: ATP γ S (adenosine 5'-O-(3-thiotriphosphate)) was purchased from Roche (Cat. No. 11162306001).

We added 10% ADP to check whether contaminant ADP might be influencing ATP or ATP γ S data. Remarkably, adding ADP if anything exaggerates the difference between ATP and ATP γ S. We discuss why this is.

Line 301: “.. Relatedly, rebinding of ADP, as explored in our ADP supplementation experiments (Fig. 5), will tend to trigger backwards translocations. Binding of ADP to the 1HB AA state will tend to trigger a backslip or detachment (Fig. 6d). Binding of ADP to the apo forward head of a 2HB wait state would, by contrast, generate extra 8 nm mechanical backsteps (Fig. 6e).”

3) Fig. 1d-g: the y-axes on the left side should display the displacement of the bead, since displacement is the direct measurement. The load calculation, which includes uncertainty due to errors in determining trap stiffness and deviations from linearity, should be moved to the right side of the y-axis.

Done.

4) Fig. 2: the color for > 12 nm backward steps appear to be brown, not magenta.

Fixed.

5) Fig. 2i-l: are the data for 2 and 3 pN (before reaching the trigger point) the same as those in Fig. 2a and c? The authors should distinguish between data with and without rapid movement of the stage.

We have adjusted the figures and legends to address this point. Panels for which the data collection used a stage step are labelled as such on the panel. The data have been regrouped to indicate that new fig. 4 is a re-analysis using a cumulative frequency approach to obtain characteristic dwell times per force bin for the data in Fig. 3. Component analyses are provided as Supp. Figs. 1-6.

6) Fig. 2j,l: typical traces of 1 mM ATP γ S under superstall forces should be shown.

Additional superstall traces for 1 mM ATP γ S are now provided in Figure S7.

Are there successive 8 nm backward steps (Fig.1e appears to shows successive (processive) backsteps)?

Indeed there are, but not just in superstall – at all loads. These runs of 8 nm backsteps (with very few larger backwards displacements) are nearly absent in 1 μ M ATP γ S. This difference is critical and we now emphasise it more heavily in our results and discussion.

Para beginning line 263: “How does 1 mM ATP γ S generate proportionally more backsteps than 1 μ M ATP γ S?”

7) Fig. 4a-d: the populations of longer dwell times deviated from the fit. Did the authors attempt to fit with a double exponential?

Indeed we did, but the improvement was modest. We now show both single and double exponential fits in Supp. Figs 1-6.

8) Fig. 4a-d: what is the definition of λ (τ might be common)? This has been written only in the method section but should be clarified in the text or the figure legend.

R2 made the same comment. We have amended the text and figure legends to clarify.

Results line 137: "The characteristic dwell time τ , the inverse of the fitted decay constant, is obtained .."

Methods line 448: "The mean dwell time τ was calculated as the inverse of the decay constant ($1/\lambda$) and reported in seconds for each force bin."

9) Fig. 4e-h: what is the concentration of the nucleotides?

Was given in Methods, now given in fig legend also (line 500).

10) Fig. 4e-h: y-axis label is missing.

Fixed

11) Figs.3,4: the authors displayed only two examples of the histograms in Fig. 4. All histogram data with fit used to determine the mean dwell times in Figs. 3 and 4 should be provided as supplementary data or at a repository site to evaluate the accuracy of the mean dwell times.

Thank you for this comment. We now provide these data as supp. figs 1-6. Raw data are provided via FigShare.

Reviewer #4 (Remarks to the Author):

We thank R4 for their contribution.

Karnawat et al RESPONSE TO ROUND #2 REVIEWER COMMENTS

Reviewer #1 (Remarks to the Author):

The authors have made significant revisions to the manuscript, leading to considerable improvements. This work presents intriguing results in the field and is suitable for publication in Nature Communications.

Minor Comment:

The forward-to-backward probability ratios shown in Figure 2 seem to be derived from the trajectories of beads trapped in optical tweezers, such as those presented in Figures 1d-g. It would be helpful to describe this explicitly.

We thank R1 for thoughtful comments. Indeed the Fig. 2 F/B ratios refer to optically-trapped beads, as indeed do all of the data figures. We have adjusted the legend to Fig. 2 as requested.

Reviewer #2 (Remarks to the Author):

The authors have added additional experiments and analysis, particularly the ADP results, and they have clarified the point about backsteps in ATP γ S versus backslips in ATP. The new movies and movie stills are also a great addition. As I noted before, the data are very interesting and important to the field, the experiments are carried out with high technical sophistication. This is an important contribution to kinesin mechanochemistry 2025, and it builds on and extends the Tokelis 2020 Biophys J work in a nice way.

Thank you.

With that being said, I strongly feel that the entire data set can be interpreted without introducing a new Awaiting Isomerization state. I brought this up in my first review and the authors wrote a new paragraph in the discussion addressing it. The added text was a fair representation of my argument. However, I remain unconvinced and I think it is a very important point for the field and so I'm going to expand my argument here and try to bring the authors to my side. This is their paper, so the final decision is theirs.

R2 asks us to consider a simpler scheme but leaves us the option to retain our own scheme.

In my formulation, instead of a new AI state, the CLOSED state is partitioned into pre-hydrolysis state in which there is partial neck linker docking and a post-hydrolysis state where there is full neck linker docking. ATP γ S sticks the cycle in the pre-hydrolysis state for a half a second or so, solely because it is a slowly hydrolyzing analog and not due to the sulfur having any structural effect. (To me it is much more satisfying to just call it a slowly hydrolyzing analog than to implicate a structural change that has no evidence to date supporting it). The backsteps in ATP γ S result

from the conformation of this 'partial neck linker docking' (which I concede is not structurally validated, but neither is the AI state) – backsteps and sidesteps are enabled in this pre-hydrolysis CLOSED state in a way that they are not in the OPEN, apo state. Hindering loads from the trap bias the backward binding over lateral. I build the model in more detail below.

The key question differentiating the models is: what triggers the forward step from the 1HB state: ATP binding or ATP hydrolysis by the Mt-bound head? Historically, the idea that ATP binding alone triggers the step came from the Rice 1999 neck linker docking work and Hackney and Ma and Taylor half-site release experiment that found that AMPPNP and ATP γ S both trigger mADP release in the tethered head (albeit at slower rates than ATP). Later work by Milic et al (eLife 2015) found that under assisting loads, Pi could enhance the run length, consistent with dissociation occurring from the ADP state following Pi release and before the tethered head binds to the next site- this also is a feature of the Karnawat model here. The arguments for ATP hydrolysis versus ATP binding being the trigger for the step are laid out on pp1220-1222 of Hancock, BJ (2015) 110:1216–1225.

In the Tokelis 2020 BJ paper, both pathways (ATP or ADP-Pi) triggering the forward step were included, and that is continued here. It is my opinion that the bulk of the evidence supports hydrolysis being the trigger, and in my opinion the current study only reinforces this view and provides new quantitative constraints.

Using Fig 6 as a starting point, I rearranged things to show my proposed model. The differences are that the backstep starts from the closed pre-hydrolysis state and the forward step only starts from the closed post-hydrolysis state. I then define probabilities at the key 'kinetic race' transition points, and then probabilities of the different outcomes. Finally, I put numbers in to try to connect the model to the experimental data.

Key relationships:

We calculate probabilities at the important bifurcations in the model:

$$p_{BackStep} = \frac{k_{BackStep}(F)}{k_{BackStep}(F) + k_{hyd}}$$

$$p_{ForStep} = \frac{k_{ForStep}(F)}{k_{ForStep}(F) + k_{off}^{Pi}}$$

$$p_{BackSlip} = \frac{k_{BackSlip}(F)}{k_{BackSlip}(F) + k_{off}^D}$$

From these probabilities at the key bifurcation points, we can formulate probabilities of the end results (the different paths) as follows:

$$Prob(BackStep) = p_{BackStep}$$

$$Prob(ForStep) = (1 - p_{BackStep})p_{ForStep}$$

$$Prob(BackSlip) = (1 - p_{BackStep})(1 - p_{ForStep})p_{BackSlip}$$

$$Prob(NoStep) = (1 - p_{BackStep})(1 - p_{ForStep})(1 - p_{BackSlip})$$

To analytically derive the F/B ratio, first we will consider only slip probability:

$$\frac{F}{B} = \frac{(1 - p_{BackStep})p_{ForStep}}{(1 - p_{BackStep})(1 - p_{ForStep})p_{BackSlip}}$$

To simplify, assume that in ATP, $p_{BackStep} = 0$ (you only get forward steps and backslips), and assume no futile cycles, so $p_{BackSlip} = 1$. In this way we don't need to consider $k_{BackSlip}(F)$. This can be relaxed later if need be. In this case:

$$\frac{F}{B} = \frac{p_{ForStep}}{(1 - p_{ForStep})} = \frac{\frac{k_{ForStep}(F)}{k_{ForStep}(F) + k_{off}^{Pi}}}{\frac{k_{off}^{Pi}}{k_{ForStep}(F) + k_{off}^{Pi}}} = \frac{k_{ForStep}(F)}{k_{off}^{Pi}}$$

Incorporating rates from data:

- 1) From motility, overall unloaded stepping rate 100 s^{-1} in ATP and from motility and ATPase, overall unloaded stepping rate in ATPgS is $2\text{-}3 \text{ s}^{-1}$.
- 2) Let's say $k_{hyd}^{ATP} = 300 \text{ s}^{-1}$ for purpose of argument. Say $k_{hyd}^{ATPgS} = 2 \text{ s}^{-1}$ (based on AI duration is 0.5 s).
- 3) At 2-6 pN load in 1 mM ATPgS, get ~20% backsteps, 60-70% forsteps (Fig. 2D), so say $p_{for}/p_{back}=3$. From this, let's posit that $k_{backstep}(F)$ is relatively independent of force in 2-6 pN range – it's just the pulling back that enables it to happen, more force doesn't help. Thus, in 1 mM ATPgS at 2-6 pN:

$$\frac{p_{for}}{p_{back}} = \frac{k_{hyd}}{k_{BackStep}(F)} = 3$$

That sets $k_{BackStep}(F) = 0.7 \text{ s}^{-1}$ for 2-6 pN loads. This holds for both ATP and ATPgS.

Note that in 1 mM ATP,

$$\frac{p_{for}}{p_{back}} = \frac{k_{hyd}}{k_{BackStep}(F)} = \frac{300 \text{ s}^{-1}}{0.7 \text{ s}^{-1}} \sim 400$$

So very few backsteps in ATP (you see backslips instead).

- 4) Next, let's estimate parameters for back slipping in 1 mM ATP. We get this from the F/B ratio in Fig. 2C. $F/B = 714e^{-0.78 \cdot F}$. From probabilities and simplifying assumptions above,

$$\frac{F}{B} = \frac{k_{ForStep}(F)}{k_{off}^{Pi}}$$

To make it simple, let's assume that $k_{off}^{Pi} = 3.5 \text{ s}^{-1}$. That gives

$$k_{ForStep}(F) = 200e^{-0.78 \cdot F}$$

In that way we can reproduce Fig. 2C. Also, $k_B/dx = 0.78$, so distance parameter $dx = 5.2 \text{ nm}$.

5) In 1 mM ATP, the F/B ratio goes to ~3 at 6 pN, matching that of 1 mM ATP γ S. At higher loads it gets even smaller (see figs reproduced below). Because the pathways after hydrolysis are the same for ATP vs ATP γ S, this results in the For/Back ratio starting to plummet above 7 pN for ATP γ S.

1 mM ATP at left, 1 mM ATP γ S at right

6) What about the ATP γ S triggered half-site of 30/s in Mickolajczyk 2015 PNAS and Ma and Taylor 1997 JBC? This must mean that in ATP γ S in the absence of load, you can bypass hydrolysis to bind the tethered head and release ADP. In the Karnawat model in Fig 6, this pathway isn't actually there kinetically, unless perhaps you say that backstepping is slow in presence of load and sidestepping is nil whereas in the absence of load, sidestepping occurs at ~30 s⁻¹.

We think our scheme can account for rapid half-site release in response to ATP γ S, because ATP γ S-induced tethered head attachment can be fast, whilst subsequent front head detachment (which is the point at which we detect a backstep), can be slow. This proposed sequence is illustrated in Fig. 6 and in our animations.

We could also do it the following way. Say k_{hyd} is 300 and $k_{forstep}$ from the T state is 30/s. So in ATP, 10% of forward steps come before hydrolysis, but in ATP γ S, essentially all steps come before hydrolysis at the rate of 30 s⁻¹. That gives you the 30 s⁻¹ half-site in ATP γ S, but then why isn't kinesin extremely processive in ATP γ S since it never enters a vulnerable state? My opinion is that in ATP γ S in the absence of load, there is some off-pathway such as sidestepping. This would explain both the chaotic sidesteps in Mickolajczyk 2015 PNAS and would also reconcile the stopped flow half-site results. It would also apply to AMPPNP, which also triggers half-site at 32 s⁻¹ (Ma and Taylor 1997 JBC).

AMPPNP binding generates a CLOSED state that docks the NL and does a forward halfstep. This is quite different to the action of ATP γ S, which as shown by Mickolajczyk 2015, generates backsteps and off-axis steps. We believe our scheme can account for both actions, because the initial T-state in ATP γ S (the AI state) is competent to step (its tethered head is unparked), but its stepping is unsteered because it cannot dock the NL. In AMPPNP, we envisage that the AI state will be only very transiently occupied, allowing a fast transition into a CLOSED state with coupled NL docking and a forward half-step.

So, to reiterate, I think that this simpler model I propose is sufficient to explain all of the data here. This is the authors' manuscript and so it is their decision of how they want to use this information. If Nature Communications would allow me to submit a small companion piece with my proposed model, I would be open to that (assuming they waive the exorbitant

publication charges). And if Cross and colleagues would like to discuss this further, I would be happy to.

Perhaps a *Matters Arising* would be appropriate?

Whilst Will's scheme and our scheme are superficially similar (both invoke load-dependent competition between forward stepping and backslipping), there are in our view 3 important differences:

- 1. Initial T state.** In Will's scheme, ATP binding directly generates a hydrolytically-competent CLOSED state with its NL partially docked. Exit from this initial T state via hydrolysis is fast (300 s^{-1}), irreversible and load-independent. By contrast in our scheme, the initial T state (the AI state) has its NL fully undocked and is hydrolytically incompetent. Exit from the AI state establishes hydrolytic competence. The AI state is in equilibrium with the CLOSED state and the position of this equilibrium is load-dependent via NL docking.
- 2. Load dependence of hydrolysis.** In our scheme, hydrolysis depends on NL docking. In Will's scheme, NL docking depends on hydrolysis. In our scheme, hydrolysis slows down under load, because full NL docking is required to fully stabilise the hydrolytically-competent CLOSED conformation. We propose this as a core principle of the stepping mechanism, ensuring tight coupling of forward stepping to hydrolysis under load.
- 3. Pi release.** In Will's scheme, hydrolysis is fast and load-independent and the flux into the orange backslipping pathway is restricted by setting Pi release at 3.5 s^{-1} . At higher loads, there are fewer forward steps and more backslips, but the backslip dwell times will continue to be set by Pi release. Backslip dwell times cannot be less than $\sim 300 \text{ ms}$. By contrast in our scheme, the rate of backslipping is set not by Pi release, but by load-dependent hydrolysis. We have shown that backslip dwell times at substall loads in ATP depend exponentially on the load (**Fig. 3a**). At 2 pN in 1 mM ATP the average is 20-30 ms. Above stall, the backslip dwell time does plateau at around 300 ms, indicating a small but finite ATPase activity of the trail head, even with the NL fully undocked.

Can Will's simpler scheme, without the extra pre-hydrolysis T state, explain our data? We believe it cannot account for the effects of load on backslip dwell times. We therefore prefer to retain our existing scheme.

Minor comments:

Line 54: ...NL of the *leading* head, right? (Or call it bound head?)

Thank you. Changed to 'bound head'

Line 66-68: I don't agree that AMPPNP and ATP γ S differ in their propensity to adopt a CLOSED state. The key with AMPPNP is that the motor completes a step into the 2H and it gets stuck there with AMPPNP trapped in the trailing head (Guydosh and Block; Schnapp and Sheetz PNAS 1990). ATP γ S doesn't get stuck because hydrolysis eventually occurs. *"in slowing entry to the CLOSED state"* removed. The sentence works better without it.

In fig 4, the 1-CDF should be normalized to 1 for all. The different counts (which is immaterial) shift the curves from each other, but what you're trying to show is the fall oV, which is then obscured by the different amplitudes. Furthermore, my strong preference would be semi-log y rather than loglog because single exponential lines are straight and you can see the slope, which is the key parameter. I'm guessing that because of large range of durations that all data were packed against the y-axis in semilog, but those all fit the line well anyway, so not much information there.

We tried both these suggestions. The normalisation of 1-CDF to 100% is indeed an improvement, one can better compare the forms of the distributions). The suggested linearisation works less well, for the reason given by the reviewer. We have normalized the Y-axis for all the plots in the main text.

In fig 4, it would be more informative to pull out a distance parameter, dx, using the Bell equation $k(F) = k_0 * e^{(F * dx / kT)}$ rather having the exponent value be in inverse pN.

This distance parameter is readily calculable, but it remains uncertain and somewhat controversial how it relates to the stepping mechanism, so we prefer not to introduce it at this point.

Line 208: *“ATPyS binding to the MT-attached head also triggers MT-activated halfsite ADP release, as detected using Mant-ADP, at ~30 s⁻¹, much faster than the ~2 s⁻¹ rate of ATPyS-driven stepping (21,12,17) and the ~3 s⁻¹ rate of MT-activated ATPyS turnover (Fig. 1).”*

In Mickolajczyk Ref 21, we argued that the ~30 s⁻¹ ATPyS triggered mADP release was due to sidestepping. As the authors point out, the highly variable stepping traces in ATPyS were consistent with this. I think this is still the most plausible explanation. I note that in the authors' model in Fig 6, there is not a prediction for a 30 s⁻¹ ATPyS-triggered mADP release rate unless I'm mistaken. (It would go as the backstepping rate, which is informed by the dwell times.)

Will asks how slow ATPyS backstepping (or sidestepping) can be consistent with 30 s⁻¹ halfsite release in solution. We think it can, because ATPyS-induced halfsite ADP release from the backstepping tethered head can be fast, whilst subsequent detachment of the lead head, generating a mechanical backstep under backwards load (which is what we detect), can be much slower. We think that under load in ATPyS, the front head proportionates between a hydrolysis-incompetent ATPyS state with a fully undocked NL and a hydrolysis-competent ATPyS state with a partially docked NL. This slows down hydrolysis on the front head. This is the same point raised by R3 (how during an ATPyS backstep can the lead head detach with its NL undocked?). We think because of slow hydrolysis, the front head detaches slowly, and fails often to detach, accounting for both the slow rate of backstepping and the appreciable rate of ATPyS-driven forward stepping. This scenario predicts that serial backsteps are more probable in 1 mM ATPyS than in 1 μM ATPyS, as cartooned in **Fig. 6c**.

Ref 12 Tokelis is the ATPase result so ref should be later in sentence.

Done

Ref 17 is a review and of questionable relevance here.

Ref 17 removed from line 211 as requested but retained in line 215 as it relates structural to kinetic results

Missing key reference of Ma and Taylor JBC (1997) 272:724-730.

Fixed, thank you

Line 246: I would say time spent in pre-hydrolysis state instead of AI state. So I would call it a 'Awaiting Hydrolysis' (AH) state.

Changed to "*pre-hydrolysis state*" as requested.

Line 258: *"Our working model envisages that in ATP γ S, most backwards displacements originate from the AI state (Fig. 6, pink pathway), much earlier in the cycle than backslips (Fig.6 orange pathway). As a result, the difference between the average forward and backstep dwell times will largely disappear."*

I think there is no difference because for a forward step to complete you need ATP hydrolysis and for a backstep to complete you also need ATP hydrolysis (after the tethered head has bound to the rear site). And in ATP γ S, hydrolysis is the rate limiting step.

We have rephrased:

"Our working model envisages that ATP γ S-driven backsteps originate pre-hydrolysis (Fig. 6, pink pathway), much earlier in the cycle than backslips (Fig.6, orange pathway). In our model ATP γ S-driven backslips occur, by essentially the same mechanism that they occur in ATP. Both forward steps and backslips need to wait for the initial ATP γ S bound state to isomerise into hydrolytic competence, which adds a ~300 ms waiting time that dominates the overall dwell time at low loads. Summing the backstep and backslip dwell time populations further erodes the difference between the average forward and backward dwell times. Any residual differences between forward and backward average dwell times in ATP γ S will tend to disappear into the noise."

Line 269: I think it's just pre-hydrolysis.

The dwell time in 1 μ M ATP γ S looks just like 1 μ M ATP – that's because ATP binding is RLS and that must vary with load. It has to be, because all the other stuff after ATP binding is fast. This applies to the 2020 Tokelis paper as well, and I don't believe it was explicitly said there, but it's interesting. One way to do it is to say: subtract the 1 mM durations at each load from the 1 μ M ATP durations. That is duration of ATP waiting. That's interesting, and it is another way of showing the 1999 Nature Visscher/Schnitzer/Block load dependence of the K_M for ATP.

The ratio of the dwell times in 1 mM ATP versus 1 μ M ATP is constant under variable load. This is consistent with a scheme in which T-binding is not itself load-dependent, whereas ATP-dependent stepping and turnover are load-dependent.

Final note:

Reviewer #3 had this comment in the first round of reviews:

The authors cite the Mickolajczyk paper (PNAS 2015) regarding stepping motion in the presence of ATP γ S, but their off-axis displacements are significantly larger (10-20 nm) than the spacing between adjacent protofilaments (~5 nm), and their gold probes

are likely multivalent, causing unsynchronized binding and detachment of multiple kinesin molecules bound to the bead.

The authors of the current manuscript rightfully point out that if the gold particles were multivalent then you would expect to see erratic steps in ATP as well. I would like to add that Rev 3 is wrong that sidesteps are expected to generate ~5 nm steps because that is the inter-protofilament distance. The head is ~4 nm, the tag is 5.4 nm contour length so estimate mean ~3 nm, the streptavidin is ~4 nm diameter and the gold particle diameter is 15 nm radius (30 nm diameter but treat center of mass). If you add all of these up along with the 12.5 nm diameter of the microtubule, you get a distance of 38 nm. So to calculate the expected distance from the head shifting one protofilament, then consider a cylinder with a radius of 38 nm (as opposed to the 12.5 nm of the microtubule only). If there are 13 protofilaments, then $2\pi r/13 = 18.6$ nm. And so the 10-20 nm oV-axis displacements are exactly what you would predict. I would like to finish by saying that with current methodologies of gold nanoparticles, Qdots, or organic fluorophores using MINFLUX, that measuring steps in ATPγS is actually much easier than doing the experiments in 1 mM ATP. However, to our knowledge nobody has done the experiment and shown that there isn't erratic stepping for kinesin-1 in ATPγS. So maybe the Mickolajczyk erratic stepping will be proven wrong at some point, but it's stood for almost a decade now.

Mickolajczyk et al demonstrates that ATPγS drives sidestepping and backstepping at low loads. We see no reason to doubt these data and we believe our current data are fully consistent. However as was pointed out in the first round of reviewing, we don't ourselves have direct data on sidestepping. We therefore now only mention it as a possibility, we don't make any claims.

Reviewer #3 (Remarks to the Author):

The authors have responded carefully to the concerns I raised, but my greatest concern remains unanswered. For an 8 nm backstep to occur, two highly improbable transitions must happen consecutively: 1) the tethered head binds to the rear-binding site, not the forward-binding site, in the one-head-bound state, and 2) the front head detaches from the microtubule before the rear head in the two-head-bound state. The probability of the first transition increases as the hindering load increases, since the tethered head is moved backward by the load. However, in the presence of ATP, even when this first transition occurs, the second transition is likely prohibited, that is, the rear head detaches before the front head, causing futile ATP hydrolysis and a 0-nm step that cannot be detected by optical trapping assays (the pink pathway in Figure 6 is incorrect, as it branches and returns to the main pathway in most cases before a backstep occurs). This occurs due to gating mechanisms in the two-head-bound state, which stabilizes the front head in an open conformation via the backward-pulling neck linker and the rear head in a closed conformation via the forward-stretched neck linker, as has been demonstrated recently. Guydosh and Block (PNAS 2006) also demonstrated that AMPPNP or ADP/BeFx binds tightly to the rear head but can be released after the head moves to the front position.

Therefore, for the second transition to occur, there must be a mechanism to overcome the front and rear head gating mechanisms. However, the authors focus on explaining the first transition and do not provide a convincing explanation for the second. Starting from the AI state in Figure 6, when the tethered head binds to the rear binding site, the front head would more likely release bound ATPgS rather than hydrolyzing it, as has been demonstrated by Guydosh and Block. Then, ATPgS binds to the closed rear head, and its hydrolysis proceeds very slowly. But since the closed conformation of the rear head is more compatible with the hydrolysis reaction, the rear head would detach before the front head, causing futile ATPgS hydrolysis and a 0-nm step. The authors proposed that partial docking of the neck linker in the front head allows ATPgS hydrolysis. However, the microtubule-detachment rate would decrease substantially when the neck linker is pulled backward compared to when it is pulled forward as demonstrated by Niitani et al (2025).

I suspect contaminating ADP is essential to break the front head gating mechanism. The ATPgS used by the authors (Roche 11162306001) contains a non-negligible amount of ADP. I examined the Certificate of Analysis for five lots on their website (sigmaaldrich.com) and found that ADP ranges from 2.6-7.5% (4.3% on average). Under 1 mM ATPgS conditions, the contaminating ADP concentration is approximately 4 μ M, corresponding to an ADP binding rate of about 15 s⁻¹ (once every 60 ms), which is faster than the ATPgS turnover rate. This ADP-induced detachment of the front head can explain the 8-nm backstep observed with 1 mM ATPgS. In contrast, under 1 μ M ATPgS condition, the ADP binding rate decreases to approximately 0.15 s⁻¹, much slower than the ATPgS turnover rate. Under this condition, the rear head is more likely to detach before to the front head. Therefore, which head detaches from the microtubule first can be explained by whether detachment due to ATP or ATPgS hydrolysis or detachment promoted by contaminating and/or added ADP binding occurs first. Combining these considerations for the second transition with the load dependence of the first transition provides explanations for all the experimental results.

I believe the authors should address these two concerns, a reasonable explanation for breaking the front head gating and the potential role of contaminating ADP in breaking the front head gating, before the manuscript can be accepted.

Reviewer 3 asks us to address 2 remaining questions:

(1) how the front head of a backstepping dimer in ATPgS can detach, rather than failing to detach and causing the partner head to do a futile turnover.

We fully agree with the reviewers position, and with Niitani et al. (2025), that backward NL strain substantially reduces the MT detachment rate of the leading head compared to forward pulled configurations. In ATP γ S, our measurements show that the dwell between backsteps (which we take to be the pre-hydrolysis dwell) is ~ 0.5 s at saturating [ATP γ S]. Our proposal is that whilst the detachment probability of the lead head is indeed very low, it is sufficient to account for the observed 500 ms backstep dwell time in saturating ATP γ S. We propose that despite the backward load, there is a nonzero probability of partial NL docking (as envisaged for example by Milic et al. for the trail head under load) allowing the front head to hydrolyze ATP γ S and detach (in its K.ADP state) before the rear head, thereby generating slow 8 nm backsteps.

To make this as clear as possible, we have modified Fig. 6 by adding callouts to emphasise partial NL docking. We have adjusted the Fig. 6 legend.

(2) whether contaminating ADP can explain backstepping in ATPgS.

Contaminating ADP at a few % is certainly present in our ATPgS. R3 will have seen that to test the potential effects of ADP, we added 100 μ M ADP, in both 0.9 mM ATP and 0.9 mM ATPgS. The ADP concentration is then substantially higher than any expected contaminant level. In 0.9 mM ATP plus 100 μ M ADP we see a marked increase in detachments and backslips, but we do not see the ATP γ S-like pattern of predominantly 8 nm backsteps and strongly reduced load-dependence of F/B ratio and dwell-time. Conversely, when we supplement 0.9 mM ATP γ S with 0.1 mM ADP, we see a further increase in 8 nm backsteps at the expense of forward steps. We also see a characteristic pattern in which detachments are often preceded by an 8 nm backstep (Fig. 5f,m), in contrast to ATP + ADP, where detachments are usually preceded by a forward step (Fig. 5a,l). These experiments show that 10% ADP has distinct, substrate-dependent effects. Adding ADP to ATP does not make ATP behave like ATP γ S, and adding ADP to ATP γ S does not make ATP γ S behave like ATP. This argues strongly that ~ 3 – 5% ADP at 1 mM ATP γ S) cannot, by itself, account for the ATP γ S-specific backstepping mechanics, particularly the predominance of 8 nm backsteps (with very few larger backward displacements), the near load-independence of the F/B ratio at 1 mM ATP γ S, and the additional ~ 0.5 s load-independent dwell in ATP γ S relative to ATP. For this reason, we continue to view overpopulation of AI state by ATP γ S as the primary cause of the increased 8 nm backsteps and the flattening of the load dependence.

In sum, our proposal is that ATP or ATPgS or ADP binding to the front head is weak compared to binding to the trail head, but that it does happen, and that hydrolysis of ATPgS or ATP can also happen, at a reduced rate that we propose is due to stochastic, infrequent, partial NL docking.

Reviewer #4 (Remarks to the Author):

We thank Reviewer #4 for his or her contribution.

Will Hancock
10/28/2025

Review of Karnawat et al Nat Com Revision

The authors have added additional experiments and analysis, particularly the ADP results, and they have clarified the point about backsteps in ATPgS versus backslips in ATP. The new movies and movie stills are also a great addition. As I noted before, the data are very interesting and important to the field, the experiments are carried out with high technical sophistication. This is an important contribution to kinesin mechanochemistry 2025, and it builds on and extends the Tokelis 2020 Biophys J work in a nice way.

With that being said, I strongly feel that the entire data set can be interpreted without introducing a new Awaiting Isomerization state. I brought this up in my first review and the authors wrote a new paragraph in the discussion addressing it. The added text was a fair representation of my argument. However, I remain unconvinced and I think it is a very important point for the field and so I'm going to expand my argument here and try to bring the authors to my side. This is their paper, so the final decision is theirs.

In my formulation, instead of a new AI state, the CLOSED state is partitioned into pre-hydrolysis state in which there is partial neck linker docking and a post-hydrolysis state where there is full neck linker docking. ATPgS sticks the cycle in the pre-hydrolysis state for a half a second or so, solely because it is a slowly hydrolyzing analog and not due to the sulfur having any structural effect. (To me it is much more satisfying to just call it a slowly hydrolyzing analog than to implicate a structural change that has no evidence to date supporting it). The backsteps in ATPgS result from the conformation of this 'partial neck linker docking' (which I concede is not structurally validated, but neither is the AI state) – backsteps and sidesteps are enabled in this pre-hydrolysis CLOSED state in a way that they are not in the OPEN, apo state. Hindering loads from the trap bias the backward binding over lateral. I build the model in more detail below.

The key question differentiating the models is: what triggers the forward step from the 1HB state: ATP binding or ATP hydrolysis by the Mt-bound head? Historically, the idea that ATP binding alone triggers the step came from the Rice 1999 neck linker docking work and Hackney and Ma and Taylor half-site release experiment that found that AMPPNP and ATPgS both trigger mADP release in the tethered head (albeit at slower rates than ATP). Later work by Milic et al (eLife 2015) found that under assisting loads, Pi could enhance the run length, consistent with dissociation occurring from the ADP state following Pi release and before the tethered head binds to the next site- this also is a feature of the Karnawat model here. The arguments for ATP hydrolysis versus ATP binding being the trigger for the step are laid out on pp1220-1222 of Hancock, BJ (2015) **110**:1216–1225.

In the Tokelis 2020 BJ paper, both pathways (ATP or ADP-Pi) triggering the forward step were included, and that is continued here. It is my opinion that the bulk of the evidence supports hydrolysis being the trigger, and in my opinion the current study only reinforces this view and provides new quantitative constraints.

Using Fig 6 as a starting point, I rearranged things to show my proposed model. The differences are that the backstep starts from the closed pre-hydrolysis state and the forward step only starts from the closed post-hydrolysis state. I then define probabilities at the key 'kinetic race' transition points, and then probabilities of the different outcomes. Finally, I put numbers in to try to connect the model to the experimental data.

Key relationships:

We calculate probabilities at the important bifurcations in the model:

$$p_{BackStep} = \frac{k_{BackStep}(F)}{k_{BackStep}(F) + k_{hyd}}$$

$$p_{ForStep} = \frac{k_{ForStep}(F)}{k_{ForStep}(F) + k_{off}^{Pi}}$$

$$p_{BackSlip} = \frac{k_{BackSlip}(F)}{k_{BackSlip}(F) + k_{off}^D}$$

From these probabilities at the key bifurcation points, we can formulate probabilities of the end results (the different paths) as follows:

$$Prob(BackStep) = p_{BackStep}$$

$$Prob(ForStep) = (1 - p_{BackStep})p_{ForStep}$$

$$Prob(BackSlip) = (1 - p_{BackStep})(1 - p_{ForStep})p_{BackSlip}$$

$$Prob(NoStep) = (1 - p_{BackStep})(1 - p_{ForStep})(1 - p_{BackSlip})$$

To analytically derive the F/B ratio, first we will consider only slip probability:

$$\frac{F}{B} = \frac{(1 - p_{BackStep})p_{ForStep}}{(1 - p_{BackStep})(1 - p_{ForStep})p_{BackSlip}}$$

To simplify, assume that in ATP, $p_{BackStep} = 0$ (you only get forward steps and backslips), and assume no futile cycles, so $p_{BackSlip} = 1$. In this way we don't need to consider $k_{BackSlip}(F)$. This can be relaxed later if need be. In this case:

$$\frac{F}{B} = \frac{p_{ForStep}}{(1 - p_{ForStep})} = \frac{\frac{k_{ForStep}(F)}{k_{ForStep}(F) + k_{off}^{Pi}}}{\frac{k_{off}^{Pi}}{k_{ForStep}(F) + k_{off}^{Pi}}} = \frac{k_{ForStep}(F)}{k_{off}^{Pi}}$$

Incorporating rates from data:

- 1) From motility, overall unloaded stepping rate 100 s^{-1} in ATP and from motility and ATPase, overall unloaded stepping rate in ATPgS is $2-3 \text{ s}^{-1}$.
- 2) Let's say $k_{hyd}^{ATP} = 300 \text{ s}^{-1}$ for purpose of argument. Say $k_{hyd}^{ATPgS} = 2 \text{ s}^{-1}$ (based on AI duration is 0.5 s).
- 3) At 2-6 pN load in 1 mM ATPgS, get ~20% backsteps, 60-70% forsteps (Fig. 2D), so say $p_{for}/p_{back}=3$. From this, let's posit that $k_{backstep}(F)$ is relatively independent of force in 2-6 pN range – it's just the pulling back that enables it to happen, more force doesn't help. Thus, in 1 mM ATPgS at 2-6 pN:

$$\frac{p_{for}}{p_{back}} = \frac{k_{hyd}}{k_{BackStep}(F)} = 3$$

That sets $k_{BackStep}(F) = 0.7 \text{ s}^{-1}$ for 2-6 pN loads. This holds for both ATP and ATPgS.

Note that in 1 mM ATP,

$$\frac{p_{for}}{p_{back}} = \frac{k_{hyd}}{k_{BackStep}(F)} = \frac{300 \text{ s}^{-1}}{0.7 \text{ s}^{-1}} \sim 400$$

So very few backsteps in ATP (you see backslips instead).

- 4) Next, let's estimate parameters for back slipping in 1 mM ATP. We get this from the F/B ratio in Fig. 2C. $F/B = 714e^{-0.78 * F}$. From probabilities and simplifying assumptions above,

$$\frac{F}{B} = \frac{k_{ForStep}(F)}{k_{off}^{Pi}}$$

To make it simple, let's assume that $k_{off}^{Pi} = 3.5 \text{ s}^{-1}$. That gives

$$k_{ForStep}(F) = 200e^{-0.78 * F}$$

In that way we can reproduce Fig. 2C. Also, $k_B T/dx = 0.78$, so distance parameter $dx = 5.2 \text{ nm}$.

- 5) In 1 mM ATP, the F/B ratio goes to ~3 at 6 pN, matching that of 1 mM ATPgS. At higher loads it gets even smaller (see figs reproduced below). Because the pathways after hydrolysis are the

same for ATP vs ATPgS, this results in the For/Back ratio starting to plummet above 7 pN for ATPgS.

- 6) What about the ATPgS triggered half-site of 30/s in Mickolajczyk 2015 PNAS and Ma and Taylor 1997 JBC? This must mean that in ATPgS in the absence of load, you can bypass hydrolysis to bind the tethered head and release ADP. In the Karnawat model in Fig 6, this pathway isn't actually there kinetically, unless perhaps you say that backstepping is slow in presence of load and sidestepping is nil whereas in the absence of load, sidestepping occurs at $\sim 30 \text{ s}^{-1}$. We could also do it the following way. Say k_{hyd} is 300 and k_{forstep} from the T state is 30/s. So in ATP, 10% of forward steps come before hydrolysis, but in ATPgS, essentially all steps come before hydrolysis at the rate of 30 s^{-1} . That gives you the 30 s^{-1} half-site in ATPgS, but then why isn't kinesin extremely processive in ATPgS since it never enters a vulnerable state? My opinion is that in ATPgS in the absence of load, there is some off-pathway such as sidestepping. This would explain both the chaotic sidesteps in Mickolajczyk 2015 PNAS and would also reconcile the stopped flow half-site results. It would also apply to AMPPNP, which also triggers half-site at 32 s^{-1} (Ma and Taylor 1997 JBC).

So, to reiterate, I think that this simpler model I propose is sufficient to explain all of the data here. This is the authors' manuscript and so it is their decision of how they want to use this information. If Nature Communications would allow me to submit a small companion piece with my proposed model, I would be open to that (assuming they waive the exorbitant publication charges). And if Cross and colleagues would like to discuss this further, I would be happy to.

Minor comments:

Line 54: ...NL of the *leading* head, right? (Or call it bound head?)

Line 66-68: I don't agree that AMPPNP and ATPgS differ in their propensity to adopt a CLOSED state. The key with AMPPNP is that the motor completes a step into the 2H and it gets stuck there with AMPPNP trapped in the trailing head (Guydosh and Block; Schnapp and Sheetz PNAS 1990). ATPgS doesn't get stuck because hydrolysis eventually occurs.

In fig 4, the 1-CDF should be normalized to 1 for all. The different counts (which is immaterial) shift the curves from each other, but what you're trying to show is the fall off, which is then obscured by the different amplitudes. Furthermore, my strong preference would be semi-log y rather than log-log because single exponential lines are straight and you can see the slope, which is the key parameter. I'm guessing that because of large range of durations that all data were packed against the y-axis in semilog, but those all fit the line well anyway, so not much information there.

In fig 4, it would be more informative to pull out a distance parameter, dx, using the Bell equation $k(F) = k_0 \cdot e^{-(F \cdot dx/kT)}$ rather having the exponent value be in inverse pN.

Line 208: *“ATPγS binding to the MT-attached head also triggers MT-activated halfsite ADP release, as detected using Mant-ADP, at ~30 s⁻¹, much faster than the ~2 s⁻¹ rate of ATPγS-driven stepping (21,12,17) and the ~3 s⁻¹ rate of MT-activated ATPγS turnover (Fig. 1).”*

In Mickolajczyk Ref 21, we argued that the ~30 s⁻¹ ATPγS triggered mADP release was due to sidestepping. As the authors point out, the highly variable stepping traces in ATPγS were consistent with this. I think this is still the most plausible explanation. I note that in the authors' model in Fig 6, there is not a prediction for a 30 s⁻¹ ATPγS-triggered mADP release rate unless I'm mistaken. (It would go as the backstepping rate, which is informed by the dwell times.)

Ref 12 Tokelis is the ATPase result so ref should be later in sentence.

Ref 17 is a review and of questionable relevance here. Missing key reference of Ma and Taylor JBC (1997) 272:724-730.

Line 246: I would say time spent in pre-hydrolysis state instead of AI state. So I would call it a 'Awaiting Hydrolysis' (AH) state.

Line 258: *“Our working model envisages that in ATPγS, most backwards displacements originate from the AI state (Fig. 6, pink pathway), much earlier in the cycle than backslips (Fig.6 orange pathway). As a result, the difference between the average forward and backstep dwell times will largely disappear.”*

I think there is no difference because for a forward step to complete you need ATP hydrolysis and for a backstep to complete you also need ATP hydrolysis (after the tethered head has bound to the rear site). And in ATPγS, hydrolysis is the rate limiting step.

Line 269: I think it's just pre-hydrolysis.

The dwell time in 1 uM ATPγS looks just like 1 uM ATP – that's because ATP binding is RLS and that must vary with load. It has to be, because all the other stuff after ATP binding is fast. This applies to the 2020 Tokelis paper as well, and I don't believe it was explicitly said there, but it's interesting. One way to do it is to say: subtract the 1 mM durations at each load from the 1 uM ATP durations. That is duration of ATP waiting. That's interesting, and it is another way of showing the 1999 Nature Visscher/Schnitzer/Block load dependence of the K_M for ATP.

Final note:

Reviewer #3 had this comment in the first round of reviews:

The authors cite the Mickolajczyk paper (PNAS 2015) regarding stepping motion in the presence of ATPγS, but their off-axis displacements are significantly larger (10-20 nm) than the spacing between adjacent protofilaments (~5 nm), and their gold probes are likely multivalent, causing unsynchronized binding and detachment of multiple kinesin molecules bound to the bead.

The authors of the current manuscript rightfully point out that if the gold particles were multivalent then you would expect to see erratic steps in ATP as well. I would like to add that Rev 3 is wrong that sidesteps are expected to generate ~5 nm steps because that is the inter-protofilament distance. The head is ~4 nm, the tag is 5.4 nm contour length so estimate mean ~3 nm, the streptavidin is ~4 nm diameter and the gold particle diameter is 15 nm radius (30 nm diameter but treat center of mass). If you add all of these up along with the 12.5 nm diameter of the microtubule, you get a distance of 38 nm. So to calculate the expected distance from the head shifting one protofilament, then consider a cylinder with a radius of 38 nm (as opposed to the 12.5 nm of the microtubule only). If there are 13 protofilaments, then $2\pi r/13 = 18.6$ nm. And so the 10-20 nm off-axis displacements are exactly what you would predict. I would like to finish by saying that with current methodologies of gold nanoparticles, Qdots, or organic fluorophores using MINFLUX, that measuring steps in ATPgS is actually much easier than doing the experiments in 1 mM ATP. However, to our knowledge nobody has done the experiment and shown that there isn't erratic stepping for kinesin-1 in ATPgS. So maybe the Mickolajczyk erratic stepping will be proven wrong at some point, but it's stood for almost a decade now.